# Provably Efficient Multi-Task Meta Bandit Learning via Shared Representations

**Jiabin Lin and Shana Moothedath**
Department of Electrical and Computer Engineering
Iowa State University
jiabin@iastate.edu, mshana@iastate.edu

## Abstract

Learning-to-learn or meta-learning focuses on developing algorithms that leverage prior experience to quickly acquire new skills or adapt to novel environments. A crucial component of meta-learning is representation learning, which aims to construct data representations capable of transferring knowledge across multiple tasks—a critical advantage in data-scarce settings. We study how representation learning can improve the efficiency of bandit problems. We consider $T$ $d$-dimensional linear bandits that share a common low-dimensional linear representation. We provide provably fast, sample-efficient algorithms to address the two key problems in meta-learning: (1) learning a common set of features from multiple related bandit tasks and (2) transferring this knowledge to new, unseen bandit tasks. We validated the theoretical results through numerical experiments using real-world and synthetic datasets, comparing them against benchmark algorithms.

## 1 Introduction

The ability to transfer knowledge across tasks is essential for robust and sample-efficient inference and prediction [1]. Developing methods that can learn task representations capable of generalizing to unseen tasks has become increasingly critical in diverse applications, including deep reinforcement learning [2], bandit learning [3, 4], and natural language processing [5, 6]. Despite considerable advancements in transfer learning, the theoretical foundations of the underlying problem remain underdeveloped. Transfer learning for sequential decision-making problems is still in its early stages, requiring further exploration to address key gaps in understanding.

Meta-learning involves addressing two key challenges: (1) the *upstream* problem, which focuses on learning a shared model or representation across a set of source tasks to capture transferable knowledge, and (2) the *downstream* problem, which leverages this shared model to enable efficient adaptation and learning for a new target task, often under data-scarce conditions. *This paper addresses these challenges in linear bandit problems by proposing a unified framework that learns robust transferable representations and ensures efficient adaptation to data-scarce target tasks.*

Recently, a number of emerging works [7–11] investigated representation learning for bandits (upstream) and showed that if all tasks share a joint low-rank representation, then by leveraging such a joint representation, it is possible to learn faster than treating each task independently. The underlying idea is that since the tasks are related, we can efficiently extract a shared low-dimensional representation (feature extractor) and then apply a simple function—often a linear one—on top of this embedding [12–14]. Learning shared representations is inherently non-convex. While existing works have shown the benefits of representation learning, theoretical analyses often rely on convex relaxations and assume access to the optimal solution of the non-convex objective [7, 8, 10]. Moreover, the transferability of learned representations to new target tasks remains underexplored [9].

39th Conference on Neural Information Processing Systems (NeurIPS 2025).

In this paper, we focus on ensuring a desired level of accuracy for the learned representation trained on source bandit tasks with tight sample complexity while also proposing an approach that leverages this learned model to effectively handle new, unseen target bandit tasks. To learn the shared model, we introduce an explore-then-commit algorithm. We propose an Optimism in the Face of Uncertainty Learning (OFUL) algorithm designed to transfer the learned representation to unseen target tasks and provide a tight regret guarantee to address this gap. We also provide sample complexity bound of the target task for a meta-learned linear regression. Our contributions in this paper are fourfold.

1. We formulate the meta-learning problem for multi-task representation learning in linear bandits, where tasks share a low-dimensional (rank-$r$) representation, with the reward parameter for task $t \in [T + 1]$, $\theta_t^\star = B^\star w_t^\star$, $\theta_t^\star \in \mathbb{R}^d$ and $B^\star \in \mathbb{R}^{d \times r}$. Our objectives are twofold: (i) efficiently estimate the shared model $B^\star$ from the $T$ source tasks under tight regret and sample complexity guarantees (upstream problem) and (ii) develop an approach to transfer the learned model to a $(T + 1)^{\text{th}}$ unseen target task with limited data (downstream problem).

2. We propose an Explore-then-Commit (EtC) algorithm to solve the upstream problem. Our approach utilizes a careful spectral initialization followed by solving $T$ individual least-squares problems to estimate the reward parameters, avoiding relaxation of the non-convex problem. We prove that the EtC algorithm estimates the shared representation and reward parameters within $O(\sqrt{r/T})$. We provide the regret guarantee for the source tasks and the sample complexity bounds.

3. To transfer the learned model to a new task, we propose two approaches: (1) an OFUL algorithm that constructs a confidence set for the target task parameter $\theta_{T+1}^\star$ by leveraging the shared representation estimate $\widehat{B}$ learned from the source tasks. We provide high-probability guarantees that $\theta_{T+1}^\star$ lies within the confidence set and establish a tight regret bound of $\widetilde{O}(\sqrt{rdN})$ for the target task, for $N$ rounds. This represents a significant improvement over the standard bound of $\widetilde{O}(d\sqrt{N})$ by leveraging the shared model as $r \ll d$. (2) A linear regression estimator that learns from target task data using $\widehat{B}$ estimate. We present the sample complexity of meta-learned linear regression and show that it achieves significant sample reduction.

4. We evaluated the performance of our approach using synthetic datasets and real-world recommender datasets, Movielens and LastFM. We compared our approach against two benchmark methods: (i) a naive algorithm that solves tasks independently, and (ii) the Method-of-Moments (MoM) estimator in [1, 8, 15]. Our proposed approach consistently outperforms both benchmarks.

## 2 Related Work

Representation learning aims at learning a shared representation among various 'related yet different' tasks. Since the tasks are related, we can more efficiently extract common information rather than treating each task independently [12–14, 16]. Multi-task representation learning has been widely studied in the supervised learning context in both empirical applications [5, 6, 17–19] and theoretical studies [1, 12, 14, 15, 20, 21]. These works primarily address statistical rates and do not address the exploration challenges inherent in bandit learning scenarios. Linear bandits are among the most well-studied bandit models, with prominent applications in areas such as recommender systems [22–26]. Recently, representation learning for linear bandits has garnered significant attention, as leveraging task dependencies enables achieving lower regret bounds compared to addressing each task independently [7–10, 27, 28]. A significant advantage of representation learning is its ability to transfer learned representations to new, unseen tasks, thereby accelerating the learning process even under data-sparing settings, which is not explored in the existing literature [7, 8, 10, 28].

Solving multi-task linear bandits with shared representations is inherently a non-convex estimation problem. Previous works [7, 8] assumed that the optimal solution to a nonconvex cost function is known. This assumption is used in Lemma 2 in [8] and Lemma 1 in [7] to derive the initial results for regret analysis. These works primarily focused on regret guarantees under the assumption of a known optimal estimator to validate the effectiveness of learning representations. [10] considered a convex relaxation of the problem through trace-norm regularization (Algorithm 1). The solution to the relaxed problem may not necessarily correspond to a valid solution to the original problem. [28] proposed an alternating gradient descent and minimization algorithm for estimating the unknown reward matrix without relaxing the non-convex cost function. The episodic algorithm relies on independent and identically distributed (i.i.d.) data in both exploration and commit phases, which becomes restrictive

specifically during the commit episodes where actions are chosen greedily. Another related line of work includes low-rank bilinear bandits [29–31] and generalized linear bandits [32], which consider a single bandit setting where the reward parameter is modeled as a low-rank matrix. The single-bandit setting has also been studied in [33], which proposed a kernel-based multi-task contextual bandit framework that leverages similarities among arms to improve reward estimation. However, the theoretical guarantees in [33] rely on the assumption that the task similarity matrix is known a priori. In contrast, our paper presents a meta-learning framework for multi-task representation learning in linear bandits, where multiple distinct tasks share a low-rank representation. Notably, our approach learns the shared representation from the source tasks and utilizes this learned structure to facilitate effective adaptation to a new, unseen target task in data-scarce settings.

Building on these works, our goal is to develop a provable approach with regret and estimation guarantees for multi-task representation learning and for both source and target tasks. Our main focus is on the transferability of the learned model to an unseen target task in a data-scarce setting. Meta-learning for sequential decision-making problems has recently gained popularity [34–43]. Recently [44] studied transfer learning in linear bandit using shared representations, under the ellipsoid action set assumption. Sparse structures are employed for feature learning to accelerate the learning process in [45–48]. We present additional related work in Appendix I. To the best of our knowledge, this is the first work that addressed multi-task meta bandit learning using shared representations.

## 3 Problem Formulation

**Notations:** For positive integer $n$, the set $[n]$ denotes $\{1, 2, \cdots, n\}$. For vector $x$, $\|x\|$ represents the $\ell_2$ norm and $|x|$ indicates the element-wise absolute value. For any matrix $A$, $\|A\|$ denotes the 2-norm and $\|A\|_F$ denotes the Frobenius norm. The symbol $\top$ represents the transpose of a matrix or vector. The notation $I_k$ (or sometimes just $I$) represents the $k \times k$ identity matrix, while $e_k$ denotes the $k-$th canonical basis vector. For basis matrices $B_1$ and $B_2$, we define Subspace Distance (SD) as $\mathrm{SD}(B_1, B_2) := \|(I - B_1 B_1^\top) B_2\|$. We use w.p. for with probability.

**Multi-task representation learning in linear bandits:** Let $t \in [T]$ be the index of the $T$ source tasks, and index $T + 1$ denotes the target task. Each task $t \in [T]$ addresses a related but distinct linear bandit problem. Let $\mathcal{X} \subseteq \mathbb{R}^d$ denote the finite action set. In each round $n \in [N]$, every task $t \in [T]$ independently chooses an action $x_{n,t} \in \mathcal{X}$. The task $t$ receives a corresponding reward $y_{n,t}$ from the environment, determined by the unknown but fixed reward function $y_{n,t} = x_{n,t}^\top \theta_t^\star + \eta_{n,t}$, where $\theta_t^\star$ is the unknown reward parameter and $\eta_{n,t}$ denotes noise. The expected reward is defined as $r_{n,t} = x_{n,t}^\top \theta_t^\star$, where $r_{n,t} = \mathbb{E}[y_{n,t}]$. We define $\Theta^\star := [\theta_1^\star \cdots \theta_T^\star]$ as the reward matrix, which is unknown. Given tasks are related, we can understand the problem as that all tasks share a joint representation. Following prior works such as [8, 20, 7, 28, 1, 15, 9], we consider that there exists an unknown global feature extractor $B^\star \in \mathbb{R}^{d \times r}$ and an underlying prediction parameters $w_t^\star$s such that $\theta_t^\star = B^\star w_t^\star$, for $t \in [T]$. Thus $\Theta^\star$ is a low-rank (rank-$r$) matrix, where $r \ll \min\{d, T\}$.

The goal of upstream learning is to find a near-accurate model for any task $t \in [T]$ via sufficient exploration under tight sample complexity, and output a well-learned representation for the downstream task. We are also interested in obtaining the regret guarantee for the source tasks given by

$$\mathcal{R}_{N,T} := \sum_{t=1}^{T} \sum_{n=1}^{N} (x_{n,t}^{\star\top} \theta_t^\star - x_{n,t}^\top \theta_t^\star). \tag{1}$$

where $x_{n,t}^\star$ is the optimal action for task $t$ in round $n$. Let $N_1$ denote the exploration horizon in the upstream problem. Thus the representation learning reduces to obtaining the estimates $\widehat{\Theta} = \widehat{B}\widehat{W}$ with the goal of minimizing the cost function $f(\widehat{B}, \widehat{W})$

$$f(\widehat{B}, \widehat{W}) = \sum_{n=1}^{N_1} \sum_{t=1}^{T} \|y_{n,t} - x_{n,t}^\top \widehat{B} \widehat{w}_t\|^2, \tag{2}$$

where $\widehat{B} \in \mathbb{R}^{d \times r}$ and $\widehat{W} \in \mathbb{R}^{r \times T}$. The cost function in Eq. (2) is non-convex, thus challenging to solve. Let $\Theta^\star \overset{\mathrm{SVD}}{=} B^\star \Sigma V^\star := B^\star W^\star$, $B^\star \in \mathbb{R}^{d \times r}$, $\Sigma \in \mathbb{R}^{r \times r}$, and $V^\star \in \mathbb{R}^{r \times T}$, denote (rank $r$) singular value decomposition. Thus $B^\star, V^{\star\top}$ are basis matrices and $W^\star := \Sigma V^\star$. The maximum and minimum singular values of $\Sigma$ are $\sigma_{\max}^\star$ and $\sigma_{\min}^\star$, respectively, and condition number $\kappa := \frac{\sigma_{\max}^\star}{\sigma_{\min}^\star}$.

**Algorithm 1** Explore-then-Commit (EtC) Algorithm for Representation Learning in Linear Bandits

---

1: **Parameters:** Total number of rounds, $N$; Number of rounds for exploration step, $N_1$; Multiplier in specifying $\alpha$ for init step, $\tilde{C} = 9\kappa^2\mu^2$; $\widehat{\theta}_t \leftarrow 0$ for all $t \in [T]$
2: **for** $n \leftarrow 1, \cdots, N_1$ **do**
3:     For every task $t \in [T]$, randomly select an action $x_{n,t}$ and observe $y_{n,t}$.
4: **end for**
5: Compute $Y_{N_1,t} = [y_{1,t}, \cdots, y_{N_1,t}]^\top$, $\Phi_{N_1,t} = [x_{1,t}, \cdots, x_{N_1,t}]^\top$ for $t \in [T]$
6: **Spectral Initialization**
7: $Y_{t,trunc}(\alpha) := Y_{N_1,t} \circ \mathbb{1}_{\{|Y_{N_1,t}| \leqslant \sqrt{\alpha}\}}$, where $\alpha = \frac{\tilde{C}}{N_1 T} \sum_{n=1,t=1}^{N_1,T} y_{n,t}^2$
8: $\widehat{\Theta}_0 := \frac{1}{N_1} \sum_{t=1}^{T} \Phi_{N_1,t}^\top Y_{t,trunc}(\alpha) e_t^\top$
9: Set $\widehat{B} \leftarrow$ top-$r$-singular-vectors of $\widehat{\Theta}_0$
10: **Update** $\widehat{w}_t, \widehat{\theta}_t$**:** For each $t \in [T]$, set $\widehat{w}_t \leftarrow (\Phi_{N_1,t}\widehat{B})^\dagger Y_{N_1,t}$ and set $\widehat{\theta}_t = \widehat{B}\widehat{w}_t$
11: **for** $n \leftarrow N_1 + 1, \cdots, N$ **do**
12:     For each task $t \in [T]$: choose action $x_{n,t} = \arg\max_{x \in \mathcal{X}} x^\top \widehat{\theta}_t$, and obtain $y_{n,t}$
13: **end for**

---

**Transfer learning in linear bandits:** In the transfer (downstream) learning setting, the agent is assigned a new unseen target task $T + 1$. Let $N_2$ denote the learning horizon of the target task. During rounds $n \in [N_2]$, the target task selects an action $x_{n,T+1} \in \mathcal{X}$, and receives a reward $y_{n,T+1} = x_{n,T+1}^\top \theta_{T+1}^\star + \eta_{n,T+1}$. The target task shares the same feature extractor $B^\star$ with the source tasks, specifically $\theta_{T+1}^\star = B^\star w_{T+1}^\star$. The objective of the target task is to utilize the common feature extractor learned from the source tasks to more accurately estimate its own parameter $\theta_{T+1}^\star$, i.e., to minimize the (pseudo) regret of the target task

$$\mathcal{R}_{N_2,T+1} := \sum_{n=1}^{N_2} x_{n,T+1}^{\star^\top} \theta_{T+1}^\star - \sum_{n=1}^{N_2} x_{n,T+1}^\top \theta_{T+1}^\star.$$

**Other assumptions:** We now present the other assumptions used in our theoretical analysis.

**Assumption 3.1** (Distribution of Feature Vectors and Noise)**.** We assume that for every source task $t \in [T]$, the feature vector $x_{n,t}$ follows a standard Gaussian distribution. The noise $\eta_{n,t}$ is assumed to be i.i.d. Gaussian with zero mean and variance $\sigma^2$.

**Assumption 3.2** (Bounded Norm of Task Parameter)**.** We assume the existence of constants $l$ and $u$, where $0 < l \leqslant u$ such that $l \leqslant \|w_t^\star\|_2 \leqslant u$ for all $t \in [T]$.

Assumption 3.2 implies column-wise incoherence of the true reward matrix $\Theta^\star$—elaborated in Appendix A. This is critical for interpolating across columns based on localized observations $y_{n,t}$ that depend only on individual columns of $\Theta^\star$. Incoherence of the ground-truth matrices is a key property required for efficient matrix estimation and other sensing problems with sparse measurements [49, 50] and has been used in recent theoretical works on representation learning [1, 15, 21]. Assumption 3.1 is utilized in obtaining the estimation guarantees for $\widehat{B}$ using spectral initialization. We note that Assumption 3.1 applies to the source tasks but not to the transfer learning for the target task. Relaxing the Gaussian model on source task features and noise is a part of our future work.

# 4 Multi-Task Representation Learning for Linear Bandits

## 4.1 Proposed Explore-then-Commit Algorithm

This section introduces our proposed Explore-then-Commit (EtC) algorithm for multi-task representation learning in linear bandits. Our algorithm consists of two phases: an exploration phase and a commit phase. During the exploration phase, the algorithm collects data by exploring the action space. The goal is to gather sufficient information, using as few samples as possible, to estimate the shared feature extractor. Based on the knowledge obtained during the exploration phase, the algorithm estimates the unknown parameters and commits to a fixed or near-optimal strategy (policy or model) for subsequent decisions. The pseudocode of the algorithm is given in Algorithm 1.

**Exploration and spectral initialization for estimating** $(\widehat{B}, \widehat{W})$**:** In the exploration phase, for each round and task, $n \in [N_1]$ and $t \in [T]$, actions $x_{n,t}$ are chosen randomly. After exploration, our

proposed algorithm estimates the shared feature extractor and reward parameters $\widehat{\Theta} = \widehat{B}\widehat{W}$ by minimizing the cost function $f(\widehat{B}, \widehat{W})$ in Eq. (2). Due to the non-convex nature of $f(\widehat{B}, \widehat{W})$, we implement a spectral initialization to estimate $\widehat{B}$ and subsequently use the least squares estimator to estimate $\widehat{w}_t$ for each task $t \in [T]$ separately. Our goal is to use as few samples (exploration rounds) as possible. The Method-of-Moments (MoM) estimator in [1] does not bound $\|\widehat{\Theta} - \Theta^\star\|$ under the desired sample complexity. We provide a detailed explanation in Appendix B. We also demonstrate the effectiveness of our approach as compared to MoM-based approach through simulations. We address this by borrowing the truncation idea from the phase retrieval literature [49, 51, 28]. Spectral initialization was employed in [28] for initializing the alternating gradient descent and minimization estimator. While we utilize spectral initialize, we do not employ the alternating approach from [28]. Define the data matrices $Y_{N_1,t} := [y_{1,t}, \cdots, y_{N_1,t}]^\top$ and $\Phi_{N_1,t} := [x_{1,t}, \cdots, x_{N_1,t}]^\top$, for $t \in [T]$. Using the proposed spectral initialization, we define $\widehat{B}$ as the top-$r$ singular vectors of

$$\widehat{\Theta}_0 := \frac{1}{N_1} \sum_{t=1}^{T} \Phi_{N_1,t}^\top Y_{t,trunc}(\alpha) e_t^\top,$$

where $Y_{t,trunc}(\alpha) := Y_{N_1,t} \circ \mathbb{1}_{\{|Y_{N_1,t}| \leqslant \sqrt{\alpha}\}}$ and $\alpha = \frac{\tilde{C}}{N_1 T} \sum_{n=1,t=1}^{N_1,T} y_{n,t}^2$. Here, $\tilde{C} = 9\kappa^2 \mu^2$ is a constant. Note that the summation includes only those $n, t$ for which $y_{n,t}^2$ is not excessively large, i.e., not significantly larger than its empirically computed average. This truncation filters out outlier-like measurements, focusing on the remaining values. Theoretically, this transformation converts the summands into sub-Gaussian random variables with lighter tails compared to the untruncated counterparts, enabling us to establish the desired concentration bound. After fixing the estimate $\widehat{B}$, we perform $T$ independent least squares to estimate $\widehat{w}_t$ in Eq. (2) as given below.

$$\widehat{w}_t = (\Phi_{N_1,t}\widehat{B})^\dagger Y_{N_1,t}, \text{ for } t \in [T].$$

The estimates for $\theta_t^\star$ are given by $\widehat{\theta}_t = \widehat{B}\widehat{w}_t$. Proposition B.1 [28] provides guarantees of the subspace distance for spectral initialization; with high probability, we have $\mathrm{SD}(\widehat{B}, B^\star) \leqslant \delta_0$, for $\delta_0 < 0.1$.

**Commit phase:** Each task $t \in [T]$ uses the estimates $\widehat{\theta}_t = \widehat{B}\widehat{w}_t$ obtained from the exploration phase to greedily choose actions that maximize the expected reward. We present the guarantees below.

## 4.2 Main Results and Guarantees of Algorithm 1

We first present a bound for the estimation error $\|\widehat{B}\widehat{w}_t - B^\star w_t^\star\|$ after the exploration phase of Algorithm 1 and then bound the cumulative regret $\mathcal{R}_{N,T}$.

**Theorem 4.1.** *Assume Assumptions 3.1, 3.2 hold, and the noise-to-signal ratio* NSR $\leqslant \frac{cT\delta_0^2}{r^2\kappa^4\sigma_{\min}^{\star 2} N_1}\|\theta_t^\star\|^2$. *Pick a* $\delta_0 < 0.1$. *If* $N_1 \geqslant C\max(\log d, \log T, r)$ *and* $N_1 T \geqslant C\mu^2\kappa^4\frac{dr^2}{\delta_0^2}$, *then for each task* $t \in [T]$, *with probability at least* $1 - 6d^{-10}$, *Algorithm 1 at the end of exploration achieves*

$$\|\widehat{B}\widehat{w}_t - B^\star w_t^\star\| \leqslant \left(1.12 + \frac{c}{\kappa^2 r\sqrt{N_1}}\right)\mu\sqrt{\frac{r}{T}}\sigma_{\max}^\star\delta_0.$$

The proof of Theorem 4.1 is given in Appendix C. Under the stated assumptions and sample complexity requirements for the exploration step, we develop a high-probability upper bound on $\|\widehat{B}\widehat{w}_t - B^\star w_t^\star\|$. The total number of source samples needed for the exploration step $N_1 T$ is inversely related to $\delta_0$. Thus to achieve a smaller $\delta_0$, i.e., a tighter error bound $\|\widehat{B}\widehat{w}_t - B^\star w_t^\star\|$, a larger sample size is required. This highlights the trade-off between sample complexity and estimation accuracy.

**Remark 4.2.** Our guarantees hold when the noise-to-signal ratio (NSR) is below a threshold that depends on the number of source task samples, reflecting the increasing accuracy of the estimated representation with more data. NSR in low-rank estimation is defined as the ratio of the maximum eigenvalue of $\mathbb{E}[\mu\mu^\top] = \sum_t \mathbb{E}[\mu_t\mu_t^\top] = T\sigma^2 I$ to the minimum nonzero eigenvalue of $\Theta^\star\Theta^{\star\top}$, which is $\sigma_{\min}^{\star 2}$. This definition ensures that we are considering the ratio between the worst-case (largest) noise power in any direction to the smallest signal power in any direction. Thus NSR $:= \frac{T\sigma^2}{\sigma_{\min}^{\star 2}}$.

**Theorem 4.3.** *Assume Assumptions 3.1, 3.2 hold, and* NSR $\leqslant \frac{cT\delta_0^2}{r^2\kappa^4\sigma_{\min}^{\star 2} N_1}\|\theta_t^\star\|^2$. *Pick a* $\delta_0 < 0.1$. *If* $N_1 \geqslant C\max(\log d, \log T, r)$ *and* $N_1 T \geqslant C\mu^2\kappa^4\frac{dr^2}{\delta_0^2}$, *then for any* $\delta \in (0, 1)$, *with probability at*

**Algorithm 2** OFUL-Based Meta Bandit Learning using Shared Representations

1: Set number of rounds for target task, $N_2$; $\bar{V}_{0,T+1} = \lambda I$
2: Perform representation learning using source tasks and estimate $\widehat{B}$ using Algorithm 1 from line 1 to line 10
3: **for** $n \leftarrow 1, \cdots, N_2$ **do**
4:     Construct the confidence ellipsoid $\beta_n$ as Eq (4)
5:     Choose the action-estimate pair $(x_{n,T+1}, \tilde{\theta}_{n,T+1}) = \arg\max_{x \in \mathcal{X}, \theta \in \beta_n} x^\top \theta$
6:     Play action $x_{n,T+1}$ and receive the reward $y_{n,T+1}$
7:     Update $\bar{V}_{n,T+1} = \bar{V}_{n-1,T+1} + x_{n,T+1} x_{n,T+1}^\top$, $V_{n,T+1} = \widehat{B}^\top \bar{V}_{n,T+1} \widehat{B}$,
       $\widehat{w}_{n,T+1} = (\widehat{B}^\top \bar{V}_{n,T+1} \widehat{B})^{-1} \sum_{m=1}^n \widehat{B}^\top x_{m,T+1} y_{m,T+1}$, $\widehat{\theta}_{n,T+1} = \widehat{B} \widehat{w}_{n,T+1}$
8: **end for**

least $1 - 4\delta - 6d^{-10}$, *the cumulative regret of Algorithm 1 is bounded by*

$$\mathcal{R}_{N,T} \leqslant 2uT\sqrt{N \log\frac{1}{\delta} \log\frac{NT}{\delta}} + 4\mu\sigma_{\max}^\star \delta_0 \left(1.12 + \frac{c}{\kappa^2 r\sqrt{N_1}}\right) \sqrt{rNT \log\frac{1}{\delta} \log\frac{NT}{\delta}}.$$

### 4.3   Proof Sketch (Details in Appendices C and D)

Complete proof of Theorem 4.3 is given in Appendix D. Using spectral initialization, we have the guarantee for the estimate of the shared model $B^\star$ as $\mathrm{SD}(\widehat{B}, B^\star) \leqslant \delta_0$, for $\delta_0 < 0.1$. The least squares estimate of $W^\star$ is given by

$$\widehat{w}_t = (\widehat{B}^\top \Phi_{N_1,t}^\top \Phi_{N_1,t} \widehat{B})^{-1} (\Phi_{N_1,t} \widehat{B})^\top Y_{N_1,t}. \tag{3}$$

By substituting $Y_{N_1,t} = \Phi_{N_1,t} B^\star w_t^\star + H_{N_1,t}$, where $H_{n,t} = [\eta_{1,t} \cdots \eta_{n,t}]^\top$, we can rewrite Eq. (3)

$$\widehat{w}_t = (\widehat{B}^\top \Phi_{N_1,t}^\top \Phi_{N_1,t} \widehat{B})^{-1} \widehat{B}^\top \Phi_{N_1,t}^\top H_{N_1,t} + \widehat{B}^\top B^\star w_t^\star + (\widehat{B}^\top \Phi_{N_1,t}^\top \Phi_{N_1,t} \widehat{B})^{-1} \widehat{B}^\top \Phi_{N_1,t}^\top \Phi_{N_1,t} (I - \widehat{B}\widehat{B}^\top) B^\star w_t^\star.$$

We multiply both sides by $\widehat{B}$ and simplify further to derive

$$\widehat{B}\widehat{w}_t - B^\star w_t^\star = \widehat{B}(\widehat{B}^\top \Phi_{N_1,t}^\top \Phi_{N_1,t} \widehat{B})^{-1} \widehat{B}^\top \Phi_{N_1,t}^\top H_{N_1,t} + (\widehat{B}\widehat{B}^\top - I) B^\star w_t^\star$$
$$+ \widehat{B}(\widehat{B}^\top \Phi_{N_1,t}^\top \Phi_{N_1,t} \widehat{B})^{-1} \widehat{B}^\top \Phi_{N_1,t}^\top \Phi_{N_1,t} (I - \widehat{B}\widehat{B}^\top) B^\star w_t^\star.$$

To bound $\|\widehat{B}\widehat{w}_t - B^\star w_t^\star\|$, we utilize the Cauchy-Schwarz inequality, Proposition B.1, and Bernstein inequality. To bound the total cumulative regret $\mathcal{R}_{N,T}$ we bound the cumulative regret from exploration phase $\mathcal{R}_{N,T}^1$, and cumulative regret of commit phase $\mathcal{R}_{N,T}^2$ separately and combine these two bounds. To bound each of these terms, we use a combination of Azuma-Hoeffding inequality and the bound of $\|\widehat{B}\widehat{w}_t - B^\star w_t^\star\|$ from Theorem 4.1.

**Remark 4.4.** By Theorem 4.3, the cumulative regret is linear in the number of source tasks $T$. However, under Assumption 3.2, we demonstrate in Appendix A that the ground-truth matrix is incoherent, i.e., its column norms have similar magnitudes. Incoherence is essential since our measurement matrices are column-wise sparse. Utilizing this, we have a sublinear regret guarantee, $\mathcal{R}_{N,T} = \tilde{O}(\sqrt{rNT})$. [8] provided a lower bound for the cumulative regret under the infinite action set setting. In their scenario, the regret during the exploration phase increases linearly with $T$. This result is based on a more stringent assumption: the action set for all tasks and all steps is the same well-conditioned $d$-dimensional ellipsoids, which cover all directions nicely.

## 5   Transfer Learning in Bandits using Shared Representations

### 5.1   Proposed OFUL-Based Meta Bandit Learning Algorithm

This section presents our proposed OFUL meta-learning algorithm for linear bandits, consisting of $T$ source tasks and a target task $(T+1)$. The algorithm consists of two key phases: first, collaboratively learning a shared feature extractor $\widehat{B}$ from the $T$ source tasks; and second, leveraging $\widehat{B}$ to construct and maintain a confidence ellipsoid for the target task parameter $\theta_{T+1}^\star$. The pseudocode is provided in

Algorithm 2. In this section, we relax Assumption 3.1, which was essential in Section 4 for estimating the feature representation. Here we assume the noise $\eta_{n,T+1}$s are independent 1-sub-Gaussian random variables and $\|x_{n,T+1}\|_2 \leqslant L$, for $L > 0$, which is a standard assumption in the literature.

**Obtain estimate $\widehat{B}$ from source tasks:** We first estimate the shared feature $B^\star$ via exploration of $T$ source tasks followed by spectral initialization as described in Section 4. Using the estimate $\widehat{B}$ we construct a confidence set such that with high probability $\theta^\star_{T+1}$ lies in the confidence set.

**Construction of the confidence ellipsoid $\beta_n$ and estimate $\widehat{w}_{T+1}$:** After estimating $\widehat{B}$ from the source tasks, in each round $n \in [N_2]$, the target task $T + 1$ constructs a confidence ellipsoid $\beta_n$ that contains the unknown reward parameter $\theta^\star_{T+1}$. The target task then computes an optimistic estimate $\tilde{\theta}_{n,T+1} = \arg\max_{\theta \in \beta_n}(\max_{x \in \mathcal{X}} x^\top \theta)$ and chooses action $x_{n,T+1} = \arg\max_{x \in \mathcal{X}} x^\top \tilde{\theta}_{n,T+1}$ to maximize the reward based on $\tilde{\theta}_{n,T+1}$. Alternatively, the pair $(x_{n,T+1}, \tilde{\theta}_{n,T+1})$ is chosen as $(x_{n,T+1}, \tilde{\theta}_{n,T+1}) = \arg\max_{x \in \mathcal{X}, \theta \in \beta_n} x^\top \theta$, optimizing the expected reward. We denote the feature vector and reward as $x_{n,T+1}, y_{n,T+1}$, respectively. We use the data to update the estimated reward parameter $\widehat{\theta}_{n,T+1}$ and refine the confidence ellipsoid $\beta_n$. At each round $n \in [N_2]$, we perform $\ell^2$ least squares estimation with a regularization parameter $\lambda > 0$ on the data to estimate $\widehat{w}_{n,T+1}$ by minimizing

$$\arg\min_{w \in \mathbb{R}^r} \sum_{m=1}^{n} \|y_{m,T+1} - x_{m,T+1}^\top \widehat{B} w\|^2 + \lambda \|w\|_2^2.$$

From the least squares estimate $\widehat{w}_{n,T+1}$ and the estimate $\widehat{B}$ from source tasks, we have $\widehat{\theta}_{n,T+1} = \widehat{B}\widehat{w}_{n,T+1}$. We define $d \times d$ positive definite matrice $\bar{V}_{n,T+1} = \lambda I + \Phi_{n,T+1}^\top \Phi_{n,T+1}$ and $r \times r$ positive definite matrix $V_{n,T+1} = \widehat{B}^\top \bar{V}_{n,T+1} \widehat{B}$, where $\Phi_{n,T+1} = [x_{1,T+1}, \cdots, x_{n,T+1}]$. Using the estimate $\widehat{\theta}_{n,T+1}$, we construct a confidence ellipsoid $\beta_n$ as in Eq. (4). One of the main technical contributions in this section is the construction of a tighter confidence ellipsoid to estimate $\theta^\star_{T+1}$ in the target task. Theorem 5.1 guarantees that with high probability, $\theta^\star_{T+1} \in \beta_n$ for all $n \in [N_2]$.

## 5.2 Main Results and Guarantees for OFUL Algorithm

This section presents the main theoretical results for Algorithm 2. We provide guarantees to show that the true reward parameter $\theta^\star_{T+1}$ lies inside the confidence ellipsoid with high probability, as well as the upper bound on cumulative regret for the target task with high probability.

**Theorem 5.1.** *Assume Assumptions 3.1, 3.2 hold, $\|\theta^\star_{T+1}\|_2 \leqslant S$, and the noise-to-signal ratio* $\text{NSR} \leqslant \frac{cT\delta_0^2}{r^2\kappa^4\sigma_{\min}^{\star 2}N_1}\|\theta_t^\star\|^2$. *Pick $\delta_0 < 0.1$. If $N_1 \geqslant C\max(\log d, \log T, r)$ and $N_1 T \geqslant C\mu^2\kappa^4\frac{dr^2}{\delta_0^2}$, then for any $\delta \in (0,1)$ and $n \in [N_2]$, for the target task $T + 1$, it is guaranteed with probability at least $1 - \delta - 2d^{-10}$ that $\theta^\star_{T+1}$ is contained within the set*

$$\beta_n = \left\{ \theta \in \mathbb{R}^d : \|\widehat{\theta}_{n,T+1} - \theta\|_{\bar{V}_{n,T+1}} \leqslant \sigma\sqrt{2\log\frac{\det(V_{n,T+1})^{\frac{1}{2}}\det(\lambda I)^{-\frac{1}{2}}}{\delta}} + ((1+\delta_0)\sqrt{\lambda} + 2\sqrt{n}L\delta_0)S \right\}.$$

$$(4)$$

*Furthermore, if $N_1 T \geqslant C\mu^2\kappa^4 L^2 dr^2 N_2$, then w.p at least $1 - \delta - 2d^{-10}$, $\theta^\star_{T+1}$ is contained within*

$$\beta'_n = \left\{ \theta \in \mathbb{R}^d : \|\widehat{\theta}_{n,T+1} - \theta\|_{\bar{V}_{n,T+1}} \leqslant \sigma\sqrt{r\log\frac{1 + nL^2/\lambda}{\delta}} + \left(\sqrt{\lambda} + \frac{\sqrt{\lambda}}{\sqrt{N_2}L} + 2\sqrt{\frac{n}{N_2}}\right)S \right\}.$$

Proof of Theorem 5.1 is presented in Appendix E. Theorem 5.1 proves that, if the sample complexity conditions for the source task are satisfied, the true reward parameter for the target task $\theta^\star_{T+1}$ consistently lies within the built confidence ellipsoid $\beta_n$ with high probability. Our approach builds upon the concepts introduced in [22], which constructs a confidence ellipsoid for a single bandit problem. A comparative analysis shows that the confidence set scales with $\sqrt{d}$ in Theorem 2 [22], while our results achieve $\sqrt{r}$. Since $r \ll d$, our approach improves the bound, validating the advantages of transfer learning over the naive approach of independently learning the target task.

**Theorem 5.2.** *Assume Assumptions 3.1, 3.2 hold, $\|\theta_{T+1}^\star\|_2 \leqslant S$, and the noise-to-signal ratio* $\mathrm{NSR} \leqslant \frac{cT\delta_0^2}{r^2\kappa^4\sigma_{\min}^{\star 2}N_1}\|\theta_t^\star\|^2$. *If $N_1 \geqslant C\max(\log d, \log T, r)$ and $N_1 T \geqslant C\mu^2\kappa^4 L^2 dr^2 N_2$, then for any $\delta \in (0,1)$, w.p at least $1 - \delta - 2d^{-10}$, cumulative regret of Algorithm 2 for target task $T+1$ is*

$$\mathcal{R}_{N_2,T+1} \leqslant 2\sqrt{2dN_2\log\left(1 + \frac{N_2L^2}{\lambda}\right)} \cdot \left(\sigma\sqrt{r\log\left(\frac{1+N_2L^2/\lambda}{\delta}\right)} + (\sqrt{\lambda} + \frac{\sqrt{\lambda}}{\sqrt{N_2}L} + 2)S\right).$$

Proof of Theorem 5.2 is given in Appendix F. Theorem 5.2 shows that the bound on cumulative regret for the target task is $\tilde{O}(\sqrt{drN_2})$. Applying the method from [22] directly to learn the target bandit task will result in an $\tilde{O}(d\sqrt{N_2})$ regret. Given that $r \ll d$, our approach provides a significant improvement, validating the benefit of transfer learning.

### 5.3 Proof Sketch (Details in Appendices E and F)

Define $\bar{V}_{n,T+1} = \lambda I + \Phi_{n,T+1}^\top \Phi_{n,T+1}$ and $V_{n,T+1} = \widehat{B}^\top \bar{V}_{n,T+1}\widehat{B}$, where $\Phi_{n,T+1} = [x_{1,T+1}, \cdots, x_{n,T+1}]$. Using spectral initialization, we have $\mathrm{SD}(\widehat{B}, B^\star) \leqslant \delta_0$, for $\delta_0 < 0.1$. The least squares estimator with $\ell^2$ regularization is given by

$$\widehat{w}_{n,T+1} = V_{n,T+1}^{-1}(\Phi_{n,T+1}\widehat{B})^\top Y_{n,T+1}.$$

By substituting $Y_{n,T+1} = \Phi_{n,T+1}B^\star w_{T+1}^\star + H_{n,T+1}$, where $H_{n,t} = [\eta_{1,t} \cdots \eta_{n,t}]^\top$, we derive

$$\widehat{w}_{n,T+1} = V_{n,T+1}^{-1}\widehat{B}^\top \Phi_{n,T+1}^\top H_{n,T+1} - \lambda V_{n,T+1}^{-1}\widehat{B}^\top B^\star w_{T+1}^\star$$
$$+ \widehat{B}^\top B^\star w_{T+1}^\star + V_{n,T+1}^{-1}\widehat{B}^\top \Phi_{n,T+1}^\top \Phi_{n,T+1}(I - \widehat{B}\widehat{B}^\top)B^\star w_{T+1}^\star.$$

Consider vector $z \in \mathbb{R}^d$. By multiplying both sides by $z^\top \widehat{B}$ and utilizing the Cauchy–Schwarz inequality, we obtain

$$|z^\top(\widehat{\theta}_{n,T+1} - \theta_{T+1}^\star)| \leqslant \lambda\|\widehat{B}^\top z\|_{V_{n,T+1}^{-1}}\|\widehat{B}^\top\theta_{T+1}^\star\|_{V_{n,T+1}^{-1}} + |\widehat{B}^\top z\|_{V_{n,T+1}^{-1}}\|\widehat{B}^\top\Phi_{n,T+1}^\top H_{n,T+1}\|_{V_{n,T+1}^{-1}}$$
$$+ |z^\top(\widehat{B}\widehat{B}^\top - I)\theta_{T+1}^\star| + \|\widehat{B}^\top z\|_{V_{n,T+1}^{-1}}\|\widehat{B}^\top\Phi_{n,T+1}^\top\Phi_{n,T+1}(I - \widehat{B}\widehat{B}^\top)B^\star w_{T+1}^\star\|_{V_{n,T+1}^{-1}}.$$

By setting $z = \bar{V}_{n,T+1}(\widehat{\theta}_{n,T+1} - \theta_{T+1}^\star)$, we bound each term in the above equation. We use linear algebra concepts, similar techniques as in Theorem 1 in [22], Cauchy–Schwarz inequality, and $\mathrm{SD}(\widehat{B}, B^\star) \leqslant \delta_0$ guarantee from Section 4, to bound $\|\widehat{\theta}_{n,T+1} - \theta_{T+1}^\star\|_{\bar{V}_{n,T+1}}$. To bound regret $\mathcal{R}_{N,T+1}$, we use a combination of Cauchy-Schwarz inequality and Theorem 5.1.

**Sample Complexity of Meta-Learned Regression:** We present a meta-learned linear regression model that uses the learned $\widehat{B}$ and then estimates $\widehat{w}_{T+1}$ from $N_2$ target samples. We show that the number of target samples required is $O(\max(\log d, \log T, r))$ when using the estimate $\widehat{B}$, which is a significant reduction from direct learning (Appendix G).

## 6 Simulations

This section presents the experimental evaluation of our proposed approaches using both synthetic and real-world datasets. We considered two benchmark approaches: (i) a naive approach that independently solves $T$ tasks using the Thompson Sampling (TS) algorithm or the UCB algorithm, and (ii) MoM-based estimator from [1, 8, 15]. The MoM estimator, introduced in [1] for estimating the feature matrix, serves as a baseline for our representation learning approach and has also been utilized in [8, 15]. The MoM estimator estimates the representation matrix $\widehat{B}$ by calculating the top-$r$ singular vectors of the matrix $\widehat{\Theta} = \frac{1}{N_1T}\sum_{n=1,t=1}^{N_1,T} y_{n,t}^2 x_{n,t}x_{n,t}^\top$. The other existing approaches assume a convex relaxation technique in their simulations without learning the representation from the non-convex cost function. The naive approach serves as the performance benchmark for solving the tasks independently rather than jointly. In both the representation learning and transfer learning settings, the reward noise $\eta_{n,t}$ is sampled from a zero-mean Gaussian distribution with variance $10^{-6}$ for the representation learning and $10^{-2}$ for the transfer learning. We present some additional experiments, including a comparison with the convex relaxation approach in Appendix H.

## 6.1  Datasets

**Synthetic data:** The $B^\star \in \mathbb{R}^{d \times r}$ matrix is generated by orthonormalizing an i.i.d. standard Gaussian matrix, and $W^\star \in \mathbb{R}^{r \times T}$ is generated from an i.i.d. Gaussian distribution. The feature matrices $\Phi_{n,t}$s are generated from the standard Gaussian distribution. We set $d = 100$, $T = 100$, and $r = 2$, and alter the parameters $d$, $T$, and $r$ in the experiments to assess performance. In the transfer learning setting, we utilize $d = 100$, $T = 200$, and $r = 2$. All results are averaged over 100 independent trials. The error bars show standard errors, calculated as standard deviations divided by $\sqrt{100}$.

**Movielens:** We utilized the Movielens-100K dataset [52], which contains user ratings for movies. After pre-processing the data through collaborative filtering to address missing values, we created a rating matrix $R \in \mathbb{R}^{943 \times 1682}$ and normalized the scores from 0 to 5 by dividing by 5. We applied non-negative matrix factorization (NMF) with a latent factor dimension of $\sqrt{d}$, resulting in the factorization $R = UM$, where $U \in \mathbb{R}^{943 \times \sqrt{d}}$ and $M \in \mathbb{R}^{\sqrt{d} \times 1682}$. We consider each user as a separate task. For every task $t$, we obtain the feature vector $x_{n,t} \in \mathbb{R}^d$ by computing the outer product of the $t$-th row of $U$ and a certain column of $M$. Thus, the true reward parameter for any task $t$ is represented by the vectorized form of the identity matrix $I_{\sqrt{d}}$. Given that all tasks share a common reward parameter, the matrix $\Theta^\star$ has rank 1. We set parameters as: $d = 100$, $T = 10$, and $r = 1$.

**LastFM:** The LastFM dataset is from the online music streaming service Last.fm, including data for 1892 users and 17632 artists. We retain only those artists who have been listened to by a minimum of 30 users and only those users who have listened to at least 30 artists. For artists for whom the user has not engaged, we assign a reward of 0. We treat the listening count as the reward and subsequently normalize it to the interval $[0, 1]$, yielding a reward matrix $R \in \mathbb{R}^{741 \times 538}$. Similarly to the Movielens datasets, we utilize NMF with a latent factor dimension of $d$, resulting in the factorization $R = UM$, where $U \in \mathbb{R}^{741 \times d}$ and $M \in \mathbb{R}^{d \times 538}$. We consider each user as a separate task. For every task $t$, we formulate the feature vector $x_{n,t} \in \mathbb{R}^d$ by computing the element-wise product of the $t$-th row of $U$ and a column of $M$. The reward parameter for all tasks is specified as a vector of ones in $\mathbb{R}^d$. Thus, $\Theta^\star$ has a rank of 1. We set parameters as: $d = 100$, $T = 10$, and $r = 1$.

## 6.2  Results and Discussion

**Representation learning:** We evaluated the performance of our proposed algorithm against two benchmarks: the MoM estimator and a naive TS-based algorithm. Figure 1a presents the cumulative regret plots comparing the three algorithms for synthetic data. Figures 1d, 1e, and 1f present the cumulative regret plots for our approach after varying the rank, the number of source tasks, and the feature dimension, respectively. The figures indicate that as the dimension $d$ increases, meaning a more complex model, the cumulative regret increases. Conversely, as the number of tasks increases, indicating enhanced collaboration among tasks, the cumulative regret (summed over all tasks) increases; however, per-task cumulative regret decreases, as expected. Our plots show that increasing the number of tasks, enhancing collaboration reduces cumulative regret. In contrast, a higher rank leads to an increased cumulative regret, as expected. Figure 1b compares the performance of the proposed algorithm with respect to the benchmark algorithms for the Movielens data. Figure 1c compares the performance using LastFM. In the exploration phase of LastFM, the naive-UCB approach achieves a lower regret since it undergoes estimation in every step, whereas the proposed approach incurs regret due to its random exploration phase. However, after initialization, the proposed approach outperforms the naive method by a significant margin. Throughout all experiments, our proposed algorithm consistently outperforms the two benchmarks. Appendix H presents additional results.

**Transfer learning:** We evaluated the performance of our proposed approach for a new target task and compared it with the two benchmarks: Naive-UCB: a baseline that applies the UCB algorithm [22] directly for the target task without leveraging the source tasks, (ii) MoM-UCB: a variant of our algorithm that substitutes our spectral initialization with the MoM estimator introduced in [1]. Figures 1a, 1b , and 1c present the cumulative regret plots for the target task using synthetic data, the Movielens dataset, and the LastFM dataset, respectively, comparing the three algorithms. Our proposed algorithm consistently outperforms the two benchmark algorithms in all experiments. The naive approach presents inadequate generalization due to a lack of shared structure, whereas the MoM-based approach underperforms because the estimator cannot recover an effective representation matrix $\widehat{B}$ in limited source data.

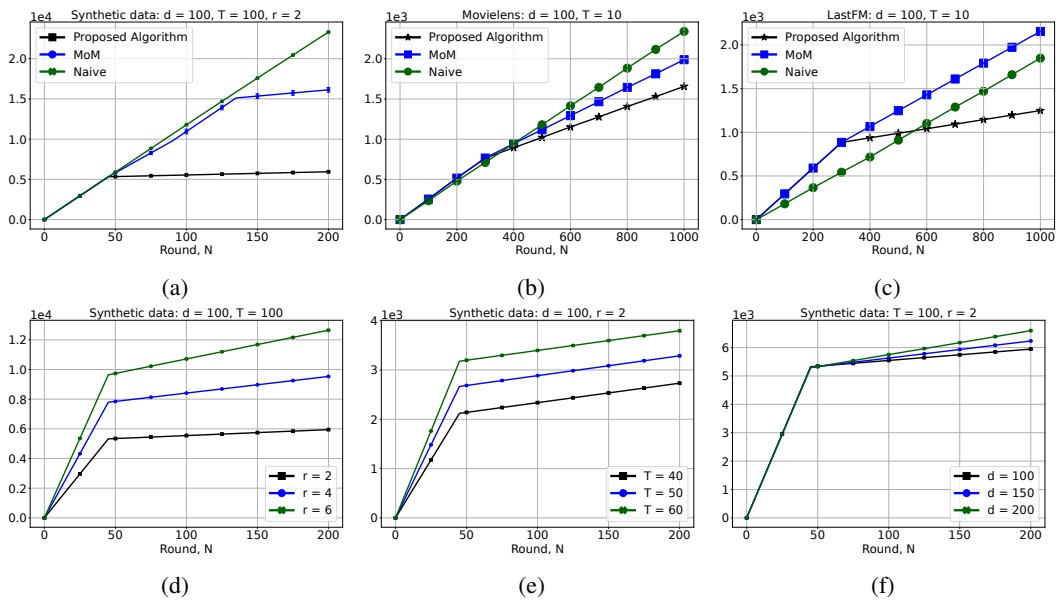

Figure 1: **Results of representation learning:** In the plots the y-axis is cumulative regret for $T$ tasks, $\mathcal{R}_{N,T}$ and x-axis is round $N$. Figures 1a, 1d, 1e, and 1f present results for synthetic data for $d = 100, T = 100, N_1 = 45$, and $N = 200$. Figures 1a, 1b, and 1c compare our proposed (EtC) algorithm against benchmark approaches (MoM and Naive) for synthetic, Movielens, and LastFM datasets. Figure 1d present plots by varying rank $r$ as $\{2, 4, 6\}$. Figure 1e presents plots varying the number of source tasks $T$ as $\{40, 50, 60\}$. Figure 1f presents plots varying the feature dimension $d$ as $\{100, 150, 200\}$. Figures 1b and 1c present the results for Movielens data and LastFM data, respectively. The parameters are set as $d = 100, T = 10, r = 1, N_1 = 300, N = 1000$.

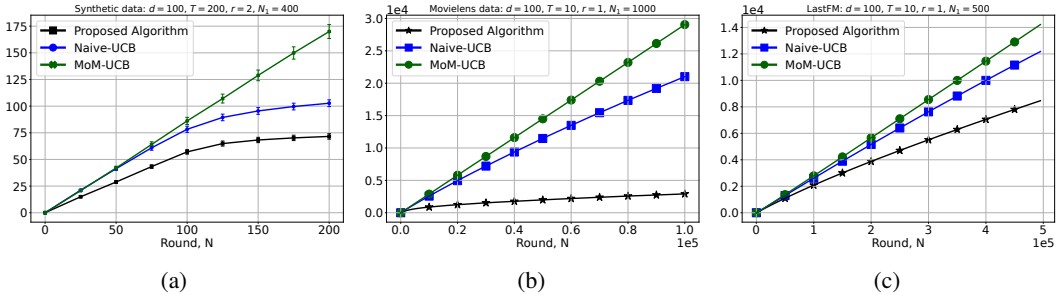

Figure 2: **Results of transfer learning:** Cumulative regret of target task vs. round. Figure 2a is for synthetic data for $d = 100, T = 200, r = 2, N_1 = 400$. Figure 2b is for Movielens data for $d = 100, T = 10, r = 1$, $N_1 = 1000$. Figure 2c is for LastFM data for $d = 100, T = 10, r = 1, N_1 = 500$.

## 7 Conclusion

This paper studied meta-learning of linear representations for linear bandits. We considered the upstream problem, which focuses on learning a shared representation from $T$ source tasks, and the downstream problem, which focuses on transferring the shared model to an unseen target task. We proposed an explore-then-commit algorithm for the upstream problem and provided convergence guarantees with regret and sample complexity bounds. Using the learned representation, we proposed an OFUL algorithm based on a confidence ellipsoid to transfer the knowledge from the source tasks to the target task. We proved the regret bound for our OFUL approach and sample complexity bound for the meta-learned regression. Finally, we evaluated the performance of our approach using synthetic data sets and two real-world data sets and compared them with benchmark approaches. As part of future work, we aim to relax Assumption 3.1 to accommodate more general feature distributions.

## 8 Acknowledgments and Disclosure of Funding

The authors thank the anonymous reviewers and the area chair whose helpful comments and suggestions helped improve the paper. This work was supported by the National Science Foundation grant NSF-CAREER 2440455.

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

# A Preliminaries

We present two concentration inequalities that are used in ur analysis in this paper.

**Proposition A.1** (Azuma-Hoeffding Inequality). *Let $\{M_j : j = 0, 1, 2, 3, \cdots\}$ be a martingale and $|M_j - M_{j-1}| \leqslant Q_j$ almost surely. Then for all positive integers $N$ and all positive reals $b$,*

$$\mathbb{P}[|M_N - M_0| \geqslant b] \leqslant \exp\left(-\frac{b^2}{2\sum_{j=1}^{N} Q_j^2}\right)$$

**Proposition A.2** (Theorem 2.8.1, [53]). *Let $X_1, \cdots, X_N$ be independent, mean zero, sub-exponential random variables. Then, for every $g \geqslant 0$, we have*

$$\mathbb{P}\left\{|\sum_{i=1}^{N} X_i| \geqslant g\right\} \leqslant 2\exp\left[-c\min\left(\frac{g^2}{\sum_{i=1}^{N} \|X_i\|_{\psi_1}^2}, \frac{g}{\max_i \|X_i\|_{\psi_1}}\right)\right],$$

*where $c > 0$ is an absolute constant.*

**Definition A.3.** (Incoherence) A rank-$r$ matrix $M \in \mathbb{R}^{d_1 \times d_2}$ is defined as $\mu$-column-wise incoherent if for every column $m_i \in \mathbb{R}^{d_1}$ of $M$, $\max_{i\in[d_2]} \|m_i\|_2 \leqslant \mu\sqrt{\frac{d_1}{d_2}}\|M\|_2$, where $\mu \geqslant 1$ is a constant that remains invariant with respect to $d_1$, $d_2$, $r$.

According to Assumption 3.2, it follows that

$$\|W^\star\|_F = \sqrt{\sum_{t=1}^{T} \|w_t^\star\|_2^2} \geqslant \sqrt{T}l.$$

Also, the Frobenius norm of $W^\star$ satisfies

$$\|W^\star\|_F = \sqrt{\sum_{i=1}^{r} \sigma_i^2(W^\star)} \leqslant \sqrt{r}\sigma_{\max}^\star.$$

Thus, by defining $\mu = \frac{u}{l} \geqslant 1$, the norm of the task parameter satisfies

$$\|w_t^\star\|_2 \leqslant u = \frac{u}{l}\frac{\sqrt{T}l}{\sqrt{T}} \leqslant \frac{u}{l}\sqrt{\frac{r}{T}}\sigma_{\max}^\star = \mu\sqrt{\frac{r}{T}}\sigma_{\max}^\star.$$

As stated in Definition A.3, the matrix $W^\star$ is $\mu$-column-wise incoherent.

**Definition A.4.** For the purpose of simplification in the demonstration, we define the matrices and vectors as follows:

- $\Phi_{n,t} := [x_{1,t} \cdots x_{n,t}]^\top$,
- $Y_{n,t} = [y_{1,t} \cdots y_{n,t}]^\top$, and
- $H_{n,t} = [\eta_{1,t} \cdots \eta_{n,t}]^\top$.

# B Spectral Initialization vs. MoM Estimator

**Spectral Initialization:** Learning the shared model is inherently a non-convex problem. In this work, we propose a solution to estimate the unknown reward matrix $\Theta^\star$ by addressing the non-convex optimization problem via a spectral initialization approach.

The standard approach used for initializing iterative algorithms for low-rank matrix estimation is to compute the top $r$ left singular vectors of the matrix

$$\widehat{\Theta}_{0,full} = \frac{1}{N_1}[\Phi_{N_1,1}^\top Y_{N_1,1}, \ldots, \Phi_{N_1,T}^\top Y_{N_1,T}]$$

$$= \frac{1}{N_1}\sum_{t=1}^{T}\sum_{n=1}^{N_1} x_{n,t}y_{n,t}e_t^\top.$$

Note that $\mathbb{E}[\widehat{\Theta}_{0,full}] = \Theta^\star$. To demonstrate the effectiveness of this initialization approach, a sin-$\Theta$ theorem, such as Davis-Kahan or Wedin, is typically employed to bound $SD(B^\star, \widehat{B})$ in terms of quantities dependent on $\widehat{\Theta}_{0,full} - \Theta^\star$. Therefore, the first requirement is to establish a bound for $\widehat{\Theta}_{0,full} - \Theta^\star$. We note that, the summands of $\widehat{\Theta}_{0,full}$ and hence of $\widehat{\Theta}_{0,full} - \Theta^\star$, are sub-exponential r.v.s. These can be bounded using the sub-exponential Bernstein inequality in Proposition A.2. This requires to bound the maximum sub-exponential norm of any summand, say we denote it as $K_m$. For our summands, we can only guarantee $K_m \leqslant (1/N_1)\max_t \|\theta_t^\star\| \leqslant (1/N_1)\mu\sqrt{r/T}\sigma_{\max}^\star$. This is not small enough, i.e., the summands are not nice enough sub-exponentials. It will require $N_1 T \succeq (d+T)r\sqrt{T}$ which is too large. To show that $\|\widehat{\Theta}_{0,full}\| \leqslant c\sigma_{\max}^\star$ with high probability under the desired sample complexity, we need $K_m$ to be of order $(r/T)$ or smaller. To achieve this we propose a truncation strategy, referred to as spectral initialization [28, 54].

Spectral initialization estimates the common feature extractor $\widehat{B}$ based on the data gathered from different tasks. Unlike the MoM approach described in Algorithm 1 of [1], our approach uses a truncation strategy to guarantee that the norm $\|\widehat{\Theta} - \Theta^\star\|$ is bounded within the desired sample complexity. We define $\widehat{B}$ as the top-$r$ singular vectors of

$$\widehat{\Theta}_0 := \frac{1}{N_1} \sum_{t=1}^{T} \Phi_{N_1,t}^\top Y_{t,trunc}(\alpha)e_t^\top,$$

where $Y_{t,trunc}(\alpha) := Y_{N_1,t} \circ \mathbb{1}_{\{|Y_{N_1,t}|\leqslant\sqrt{\alpha}\}}$ and $\alpha = \frac{\tilde{C}}{N_1 T}\sum_{n=1,t=1}^{N_1,T} y_{n,t}^2$. We present the pseudocode of the spectral initialization approach below.

We have the following guarantee from [28] for spectral initialization in linear contextual multi-task bandits. Proposition B.1 provides an error guarantee for the subspace distance between the estimated feature extractor $\widehat{B}$, obtained through spectral initialization, and the true feature extractor $B^\star$.

**Proposition B.1** (Theorem 5.1, [28]). *Assume that the noise-to-signal ratio* NSR $\leqslant \frac{cT\delta_0^2}{r^2\kappa^4\sigma_{\min}^{\star2}N_1}\|\theta_t^\star\|^2$. *Pick a $\delta_0 \leqslant 0.1$, then with probability at least $1 - \exp(\log T - cN_1) - \exp(d - \frac{c\delta_0^2 N_1 T}{r^2\mu^2\kappa^4})$, we have*

$$\mathrm{SD}(\widehat{B}, B^\star) \leqslant \delta_0.$$

**Method of Moments (MoM) Estimator:** Estimation guarantee using MoM estimator given in [1] requires the number of source task samples for each task $N \gtrsim polylog(N,d,T)(\kappa r)^4 \max(d,T)$. Our estimator, on the other hand, requires $N \gtrsim \max(\log T, \log d, r)$. Estimation guarantee in [1], Theorem 2, provides a subspace distance $\sin\theta(\widehat{B}, B^\star) = \widetilde{O}\left(\sqrt{\frac{\max(d,T)r\log N}{N}}\right)$. $\widetilde{O}(\cdot)$ here hides the logarithmic terms and constant terms. It is shown that the MoM estimator achieves close-to-optimal estimate if the number of tasks is bounded as $T \leq O(d)$. On the other hand, we provide an optimal estimation guarantee of $\delta_0$, i.e., $\mathrm{SD}(\widehat{B}, B^\star) \leqslant \delta_0$, where $\delta_0 < 0.1$, under the given sample complexity. Since our problem setting involves scenarios where the number of tasks $T$ is independent of the feature dimension $d$, the MoM estimator is not directly applicable in our setting.

## C Proof of Theorem 4.1

In this section we present the proof of Theorem 4.1. We first present the following lemma which is used in the proof of Theorem 4.1.

**Lemma C.1.** *Assume Assumption 3.1 holds. After the exploration step, for each task $t \in [T]$, with probability at least $1 - 2\exp(\log T + r - cN_1)$, we have*

$$\|\widehat{B}(\widehat{B}^\top \Phi_{N_1,t}^\top \Phi_{N_1,t}\widehat{B})^{-1}\widehat{B}^\top \Phi_{N_1,t}^\top H_{N_1,t}\| \leqslant \frac{1}{9}\sigma$$

*Proof.* To determine the upper bound for the term $\|\widehat{B}(\widehat{B}^\top \Phi_{N_1,t}^\top \Phi_{N_1,t}\widehat{B})^{-1}\widehat{B}^\top \Phi_{N_1,t}^\top H_{N_1,t}\|$, we perform a thorough analysis as follows:

$$\|\widehat{B}(\widehat{B}^\top \Phi_{N_1,t}^\top \Phi_{N_1,t}\widehat{B})^{-1}\widehat{B}^\top \Phi_{N_1,t}^\top H_{N_1,t}\| = \|(\widehat{B}^\top \Phi_{N_1,t}^\top \Phi_{N_1,t}\widehat{B})^{-1}\widehat{B}^\top \Phi_{N_1,t}^\top H_{N_1,t}\widehat{B}\|$$
$$\leqslant \|(\widehat{B}^\top \Phi_{N_1,t}^\top \Phi_{N_1,t}\widehat{B})^{-1}\|\|\widehat{B}^\top \Phi_{N_1,t}^\top H_{N_1,t}\widehat{B}\|$$

Let us consider a fixed $z \in \mathcal{S}_r$. We have

$$z^\top \widehat{B}^\top \Phi_{N_1,t}^\top \Phi_{N_1,t} \widehat{B} z = \sum_{m=1}^{N_1} z^\top \widehat{B}^\top x_{m,t} x_{m,t}^\top \widehat{B} z$$

Furthermore, we find that

$$\mathbb{E}[z^\top \widehat{B}^\top x_{m,t} x_{m,t}^\top \widehat{B} z] = z^\top \widehat{B}^\top \mathbb{E}[x_{m,t} x_{m,t}^\top] \widehat{B} z = z^\top \widehat{B}^\top \widehat{B} z = 1$$

and also

$$\mathbb{E}[z^\top \widehat{B}^\top x_{m,t}] = 0$$

$$\mathrm{Var}(z^\top \widehat{B}^\top x_{m,t}) = \mathbb{E}[z^\top \widehat{B}^\top x_{m,t} x_{m,t}^\top \widehat{B} z] = 1$$

The summands are independent sub-exponential random variables with norm $K_m \leqslant 1$. We apply sub-exponential Bernstein inequality stated in Proposition A.2 by setting $g = \epsilon_1 N_1$. In order to implement this, we show that

$$\frac{g^2}{\sum_{m=1}^{N_1} K_m^2} \geqslant \frac{\epsilon_1^2 N_1^2}{N_1} = \epsilon_1^2 N_1$$

$$\frac{g}{\max_m K_m} \geqslant \frac{\epsilon_1 N_1}{\max_m 1} = \epsilon_1 N_1$$

Therefore, for a fixed $z \in \mathcal{S}_r$, with probability at least $1 - \exp(-c\epsilon_1^2 N_1)$,

$$z^\top \widehat{B}^\top \Phi_{N_1,t}^\top \Phi_{N_1,t} \widehat{B} z - N_1 I \geqslant -\epsilon_1 N_1$$

Using epsilon-net over all $z \in \mathcal{S}_r$ adds a factor of $\exp(r)$. Thus, with probability at least $1 - \exp(r - c\epsilon_1^2 N_1)$, we have $\min_{z \in \mathcal{S}_r} z^\top \widehat{B}^\top \Phi_{N_1,t}^\top \Phi_{N_1,t} \widehat{B} z \geqslant (1 - \epsilon_1) N_1$. Then, the above holds for all $t \in [T]$ with probability at least $1 - \exp(\log T + r - c\epsilon_1^2 N_1)$. Setting $\epsilon_1 = 0.1$, we obtain

$$\|(\widehat{B}^\top \Phi_{N_1,t}^\top \Phi_{N_1,t} \widehat{B})^{-1}\| = \frac{1}{\sigma_{\min}(\widehat{B}^\top \Phi_{N_1,t}^\top \Phi_{N_1,t} \widehat{B})}$$

$$= \frac{1}{\min_{z \in \mathcal{S}_r} z^\top \widehat{B}^\top \Phi_{N_1,t}^\top \Phi_{N_1,t} \widehat{B} z}$$

$$\leqslant \frac{1}{0.9 N_1}$$

Similarly, let us consider a fixed $\bar{z} \in \mathcal{S}_r$. We have

$$\bar{z}^\top \widehat{B}^\top \Phi_{N_1,t}^\top H_{N_1,t} \widehat{B} \bar{z} = \sum_{m=1}^{N_1} \bar{z}^\top \widehat{B}^\top x_{m,t} \eta_{m,t} \widehat{B} \bar{z}$$

Furthermore, we find that

$$\mathbb{E}[\bar{z}^\top \widehat{B}^\top x_{m,t} \eta_{m,t} \widehat{B} \bar{z}] = \bar{z}^\top \widehat{B}^\top \mathbb{E}[x_{m,t} \eta_{m,t}] \widehat{B} \bar{z} = \bar{z}^\top \widehat{B}^\top \mathbb{E}[x_{m,t}] \mathbb{E}[\eta_{m,t}] \widehat{B} \bar{z} = 0$$

and also

$$\mathbb{E}[\bar{z}^\top \widehat{B}^\top x_{m,t}] = 0$$

$$\mathbb{E}[\eta_{m,t} \widehat{B} \bar{z}] = 0$$

$$\mathrm{Var}(\bar{z}^\top \widehat{B}^\top x_{m,t}) = \mathbb{E}[\bar{z}^\top \widehat{B}^\top x_{m,t} x_{m,t}^\top \widehat{B} \bar{z}] = \bar{z}^\top \widehat{B}^\top \mathbb{E}[x_{m,t} x_{m,t}^\top] \widehat{B} \bar{z} = \bar{z}^\top \widehat{B}^\top \widehat{B} \bar{z} = 1$$

$$\mathrm{Var}(\eta_{m,t} \widehat{B} \bar{z}) = \mathbb{E}[\eta_{m,t}^2 \bar{z}^\top \widehat{B}^\top \widehat{B} \bar{z}] = \mathbb{E}[\eta_{m,t}^2] \bar{z}^\top \widehat{B}^\top \widehat{B} \bar{z} = \sigma^2$$

The summands are independent sub-exponential random variables with norm $K_m \leqslant \sigma$. We apply sub-exponential Bernstein inequality stated in Proposition A.2 by setting $g = \epsilon_2 N_1 \sigma$. In order to implement this, we show that

$$\frac{g^2}{\sum_{m=1}^{N_1} K_m^2} \geqslant \frac{\epsilon_2^2 N_1^2 \sigma^2}{N_1 \sigma^2} = \epsilon_2^2 N_1$$

$$\frac{g}{\max_m K_m} \geqslant \frac{\epsilon_2 N_1 \sigma}{\sigma} = \epsilon_2 N_1$$

Therefore, for a fixed $\bar{z} \in \mathcal{S}_r$, with probability at least $1 - \exp(-c\epsilon_2^2 N_1)$,
$$\bar{z}^\top \widehat{B}^\top \Phi_{N_1,t}^\top H_{N_1,t} \widehat{B} \bar{z} \leqslant \epsilon_2 N_1 \sigma$$
Using epsilon-net over all $\bar{z} \in \mathcal{S}_r$ adds a factor of $\exp(r)$. Thus, with probability at least $1 - \exp(r - c\epsilon_2^2 N_1)$, we have $\max_{\bar{z} \in \mathcal{S}_r} \bar{z}^\top \widehat{B}^\top \Phi_{N_1,t}^\top H_{N_1,t} \widehat{B} \bar{z} \leqslant \epsilon_2 N_1 \sigma$. Then, the above holds for all $t \in [T]$ with probability at least $1 - \exp(\log T + r - c\epsilon_2^2 N_1)$. Setting $\epsilon_2 = 0.1$, we obtain
$$\|\widehat{B}^\top \Phi_{N_1,t}^\top H_{N_1,t} \widehat{B}\| = \max_{\bar{z} \in \mathcal{S}_r} \bar{z}^\top \widehat{B}^\top \Phi_{N_1,t}^\top H_{N_1,t} \widehat{B} \bar{z} \leqslant 0.1 N_1 \sigma$$
We can combine these and apply the union bound. This leads us to conclude that with probability at least $1 - 2\exp(\log T + r - cN_1)$,
$$\|\widehat{B}(\widehat{B}^\top \Phi_{N_1,t}^\top \Phi_{N_1,t} \widehat{B})^{-1} \widehat{B}^\top \Phi_{N_1,t}^\top H_{N_1,t}\| \leqslant \frac{1}{0.9 N_1} 0.1 N_1 \sigma = \frac{1}{9}\sigma$$
$\square$

**Proof of Theorem 4.1:**

We start by analyzing $\widehat{w}_t$ based on its least square estimation given by
$$
\begin{aligned}
\widehat{w}_t &= (\widehat{B}^\top \Phi_{N_1,t}^\top \Phi_{N_1,t} \widehat{B})^{-1} (\Phi_{N_1,t} \widehat{B})^\top Y_{N_1,t} \\
&= (\widehat{B}^\top \Phi_{N_1,t}^\top \Phi_{N_1,t} \widehat{B})^{-1} (\Phi_{N_1,t} \widehat{B})^\top (\Phi_{N_1,t} B^\star w_t^\star + H_{N_1,t}) \\
&= (\widehat{B}^\top \Phi_{N_1,t}^\top \Phi_{N_1,t} \widehat{B})^{-1} \widehat{B}^\top \Phi_{N_1,t}^\top H_{N_1,t} + (\widehat{B}^\top \Phi_{N_1,t}^\top \Phi_{N_1,t} \widehat{B})^{-1} \widehat{B}^\top \Phi_{N_1,t}^\top \Phi_{N_1,t} B^\star w_t^\star \\
&= (\widehat{B}^\top \Phi_{N_1,t}^\top \Phi_{N_1,t} \widehat{B})^{-1} \widehat{B}^\top \Phi_{N_1,t}^\top H_{N_1,t} + (\widehat{B}^\top \Phi_{N_1,t}^\top \Phi_{N_1,t} \widehat{B})^{-1} \widehat{B}^\top \Phi_{N_1,t}^\top \Phi_{N_1,t} \widehat{B}\widehat{B}^\top B^\star w_t^\star \\
&\quad + (\widehat{B}^\top \Phi_{N_1,t}^\top \Phi_{N_1,t} \widehat{B})^{-1} \widehat{B}^\top \Phi_{N_1,t}^\top \Phi_{N_1,t}(I - \widehat{B}\widehat{B}^\top)B^\star w_t^\star \\
&= (\widehat{B}^\top \Phi_{N_1,t}^\top \Phi_{N_1,t} \widehat{B})^{-1} \widehat{B}^\top \Phi_{N_1,t}^\top H_{N_1,t} + \widehat{B}^\top B^\star w_t^\star \\
&\quad + (\widehat{B}^\top \Phi_{N_1,t}^\top \Phi_{N_1,t} \widehat{B})^{-1} \widehat{B}^\top \Phi_{N_1,t}^\top \Phi_{N_1,t}(I - \widehat{B}\widehat{B}^\top)B^\star w_t^\star
\end{aligned}
$$
Applying $\widehat{B}$ to both sides, we derive
$$
\begin{aligned}
\widehat{B}\widehat{w}_t &= \widehat{B}(\widehat{B}^\top \Phi_{N_1,t}^\top \Phi_{N_1,t} \widehat{B})^{-1} \widehat{B}^\top \Phi_{N_1,t}^\top H_{N_1,t} + \widehat{B}\widehat{B}^\top B^\star w_t^\star \\
&\quad + \widehat{B}(\widehat{B}^\top \Phi_{N_1,t}^\top \Phi_{N_1,t} \widehat{B})^{-1} \widehat{B}^\top \Phi_{N_1,t}^\top \Phi_{N_1,t}(I - \widehat{B}\widehat{B}^\top)B^\star w_t^\star \\
&= \widehat{B}(\widehat{B}^\top \Phi_{N_1,t}^\top \Phi_{N_1,t} \widehat{B})^{-1} \widehat{B}^\top \Phi_{N_1,t}^\top H_{N_1,t} + B^\star w_t^\star + (\widehat{B}\widehat{B}^\top - I)B^\star w_t^\star \\
&\quad + \widehat{B}(\widehat{B}^\top \Phi_{N_1,t}^\top \Phi_{N_1,t} \widehat{B})^{-1} \widehat{B}^\top \Phi_{N_1,t}^\top \Phi_{N_1,t}(I - \widehat{B}\widehat{B}^\top)B^\star w_t^\star
\end{aligned}
$$
Therefore, by applying the union bound, with probability at least $1 - \exp(\log T - cN_1) - \exp(d - \frac{c\delta_0^2 N_1 T}{r^2 \mu^2 \kappa^4}) - 4\exp(\log T + r - cN_1)$, we derive
$$
\begin{aligned}
\|\widehat{B}\widehat{w}_t - B^\star w_t^\star\| &= \|\widehat{B}(\widehat{B}^\top \Phi_{N_1,t}^\top \Phi_{N_1,t} \widehat{B})^{-1} \widehat{B}^\top \Phi_{N_1,t}^\top H_{N_1,t} + (\widehat{B}\widehat{B}^\top - I)B^\star w_t^\star \\
&\quad + \widehat{B}(\widehat{B}^\top \Phi_{N_1,t}^\top \Phi_{N_1,t} \widehat{B})^{-1} \widehat{B}^\top \Phi_{N_1,t}^\top \Phi_{N_1,t}(I - \widehat{B}\widehat{B}^\top)B^\star w_t^\star\| \\
&\leqslant \|\widehat{B}(\widehat{B}^\top \Phi_{N_1,t}^\top \Phi_{N_1,t} \widehat{B})^{-1} \widehat{B}^\top \Phi_{N_1,t}^\top H_{N_1,t}\| + \|(\widehat{B}\widehat{B}^\top - I)B^\star w_t^\star\| \\
&\quad + \|\widehat{B}(\widehat{B}^\top \Phi_{N_1,t}^\top \Phi_{N_1,t} \widehat{B})^{-1} \widehat{B}^\top \Phi_{N_1,t}^\top \Phi_{N_1,t}(I - \widehat{B}\widehat{B}^\top)B^\star w_t^\star\| \\
&\leqslant \|\widehat{B}(\widehat{B}^\top \Phi_{N_1,t}^\top \Phi_{N_1,t} \widehat{B})^{-1} \widehat{B}^\top \Phi_{N_1,t}^\top H_{N_1,t}\| + \|(\widehat{B}\widehat{B}^\top - I)B^\star w_t^\star\| \\
&\quad + \|(\widehat{B}^\top \Phi_{N_1,t}^\top \Phi_{N_1,t} \widehat{B})^{-1} \widehat{B}^\top \Phi_{N_1,t}^\top \Phi_{N_1,t}(I - \widehat{B}\widehat{B}^\top)B^\star w_t^\star\| \\
&\leqslant \|\widehat{B}(\widehat{B}^\top \Phi_{N_1,t}^\top \Phi_{N_1,t} \widehat{B})^{-1} \widehat{B}^\top \Phi_{N_1,t}^\top H_{N_1,t}\| + (1 + 0.12)\|(I - \widehat{B}\widehat{B}^\top)B^\star\| \|w_t^\star\|
\end{aligned}
$$
$$\tag{5}$$
$$\leqslant \frac{1}{9}\sigma + 1.12\mu\sqrt{\frac{r}{T}}\sigma_{\max}^\star \delta_0 \tag{6}$$
$$\leqslant \frac{c}{\kappa^2 r \sqrt{N_1}}\|\theta_t^\star\|\delta_0 + 1.12\mu\sqrt{\frac{r}{T}}\sigma_{\max}^\star \delta_0 \tag{7}$$
$$\leqslant \left(1.12 + \frac{c}{\kappa^2 r \sqrt{N_1}}\right)\mu\sqrt{\frac{r}{T}}\sigma_{\max}^\star \delta_0 \tag{8}$$

where Eq (5) is derived from Proposition B.1 in [28]. Eq (6) is derived from Proposition B.1 and Lemma C.1. Eq (7) is derived from $\text{NSR} \leqslant \frac{cT\delta_0^2}{r^2\kappa^4\sigma_{\min}^{\star 2}N_1}\|\theta_t^\star\|^2$. Eq (8) is derived from $\|\theta_t^\star\| = \|w_t^\star\| \leqslant \mu\sqrt{\frac{\tau}{T}}\sigma_{\max}^\star$. To ensure probability at least $1 - 6d^{-10}$ guarantees for our theorem, it is necessary to set the bounds for $N_1$ and $N_1 T$. These bounds must guarantee that the following probability is at least $1 - 6d^{-10}$: $1 - \exp(\log T - cN_1) - \exp(d - \frac{c\delta_0^2 N_1 T}{r^2\mu^2\kappa^4}) - 4\exp(\log T + r - cN_1)$. This required that each exponential term be substantially smaller than or equal to $d^{-10}$. We obtain

$$\log T - cN_1 \leqslant -10\log d \Rightarrow N_1 \geqslant C\max(\log d, \log T)$$

$$d - \frac{c\delta_0^2 N_1 T}{r^2\mu^2\kappa^4} \leqslant -10\log d \Rightarrow N_1 T \geqslant C\mu^2\kappa^4\frac{dr^2}{\delta_0^2}$$

$$\log T + r - cN_1 \leqslant -10\log d \Rightarrow N_1 > C\max(\log d, \log T, r).$$

Consequently, combining these results, we conclude that

$$N_1 \quad \geqslant \quad C\max(\log d, \log T, r)$$
$$N_1 T \quad \geqslant \quad C\mu^2\kappa^4\frac{dr^2}{\delta_0^2}.$$

Thus, the proof is complete. $\qquad\qquad\qquad\qquad\qquad\qquad\qquad\qquad\qquad\qquad\qquad$ $\square$

## D  Proof of Theorem 4.3

**Proof of Theorem 4.3:**

We start the analysis by founding a bound on the cumulative regret $\mathcal{R}_{N,T}^1$ for the exploration step in the following manner:

$$\mathcal{R}_{N,T}^1 = \sum_{m=1}^{N_1}\sum_{t=1}^{T} x_{m,t}^{\star\top}\theta_t^\star - x_{m,t}^\top\theta_t^\star \leqslant \sum_{m=1}^{N_1}\sum_{t=1}^{T} x_{m,t}^{\star\top}\theta_t^\star$$

Let us define $M_j = \sum_{m=1}^{j}\sum_{t=1}^{T} x_{m,t}^{\star\top}\theta_t^\star$. It is observed that $\mathbb{E}[M_j|M_1,\cdots,M_{j-1}] = M_{j-1}$ and $\mathbb{E}[|M_j|] < \infty$ constitutes a martingale. According to Assumption 3.1, the feature vector $x_{m,t}^\star$ follows the standard Gaussian distribution. Thus, $x_{m,t}^{\star\top}\theta_t^\star \sim \mathcal{N}(0, \|\theta_t^\star\|^2)$. Utilizing Gaussian tail bounds and the union bound over $T$ tasks and $N_1$ rounds, with probability at least $1 - \delta$, $x_{m,t}^{\star\top}\theta_t^\star \leqslant \sqrt{2\log\frac{N_1 T}{\delta}}\|\theta_t^\star\|$. Given that

$$|M_j - M_{j-1}| = \sum_{t=1}^{T} x_{j,t}^{\star\top}\theta_t^\star$$
$$\leqslant \sum_{t=1}^{T}\sqrt{2\log\frac{N_1 T}{\delta}}\|\theta_t^\star\|$$
$$= \sum_{t=1}^{T}\sqrt{2\log\frac{N_1 T}{\delta}}\|w_t^\star\| \qquad (9)$$
$$\leqslant \sqrt{2\log\frac{N_1 T}{\delta}}uT$$

we utilize the Azuma-Hoeffding inequality stated in Proposition A.1 and the union bound to determine that with probability at least $1 - 2\delta$, the cumulative regret $\mathcal{R}_{N,T}^1$ for the exploration step is bounded as follows:

$$\mathcal{R}_{N,T}^1 = \sum_{m=1}^{N_1}\sum_{t=1}^{T}(x_{m,t}^\star - x_{m,t})^\top\theta_t^\star \leqslant 2uT\sqrt{N_1\log\frac{1}{\delta}\log\frac{N_1 T}{\delta}}.$$

According to Assumption 3.2, matrix $W^\star$ is $\mu$-column-wise incoherence (Appendix A). This indicates that the norm of each task-specific vector is bounded as $\|w_t^\star\|_2 \leqslant \mu\sqrt{\frac{\tau}{T}}\sigma_{\max}^\star$. By applying this

property into the analysis of Eq. (9), we conclude that with probability at least $1 - 2\delta$, the cumulative regret is bounded by $\mathcal{R}^1_{N,T} \leqslant 2\mu\sigma^\star_{\max}\sqrt{rN_1T\log\frac{1}{\delta}\log\frac{N_1T}{\delta}}$.

Next, we demonstrate a bound for the cumulative regret $\mathcal{R}^2_{N,T}$ in the following round as follows:

$$
\begin{aligned}
\mathcal{R}^2_{N,T} &= \sum_{m=N_1}^{N}\sum_{t=1}^{T} x^{\star\top}_{m,t}\theta^\star_t - x^\top_{m,t}\theta^\star_t \\
&= \sum_{m=N_1}^{N}\sum_{t=1}^{T} x^{\star\top}_{m,t}\theta^\star_t - x^\top_{m,t}\theta^\star_t + x^{\star\top}_{m,t}\widehat{\theta}_t - x^{\star\top}_{m,t}\widehat{\theta}_t \\
&\leqslant \sum_{m=N_1}^{N}\sum_{t=1}^{T} x^{\star\top}_{m,t}\theta^\star_t - x^\top_{m,t}\theta^\star_t + x^\top_{m,t}\widehat{\theta}_t - x^{\star\top}_{m,t}\widehat{\theta}_t \\
&= \sum_{m=N_1}^{N}\sum_{t=1}^{T} x^{\star\top}_{m,t}(\theta^\star_t - \widehat{\theta}_t) + x^\top_{m,t}(\widehat{\theta}_t - \theta^\star_t)
\end{aligned}
$$

Let us define $M'_j = \sum_{m=N_1}^{j}\sum_{t=1}^{T} x^{\star\top}_{m,t}(\theta^\star_t - \widehat{\theta}_t) + x^\top_{m,t}(\widehat{\theta}_t - \theta^\star_t)$. Observing that $\mathbb{E}[M'_j|M'_1,\cdots,M'_{j-1}] = M'_{j-1}$ and $\mathbb{E}[|M'_j|] < \infty$, it can be found that $\{M'_j : j = 0,1,2,3,\cdots\}$ constitutes a martingale as well. According to Assumption 3.1, the feature vector $x_{m,t}$ follows a standard Gaussian distribution. Thus, $x^\top(\theta^\star_t - \widehat{\theta}_t) \sim \mathcal{N}(0, \|(\theta^\star_t - \widehat{\theta}_t)\|^2)$. Utilizing Gaussian tail bounds and the union bound over $T$ tasks and $(N - N_1)$ rounds, with probability at least $1 - \delta$, $x^\top(\theta^\star_t - \widehat{\theta}_t) \leqslant \sqrt{2\log\frac{(N-N_1)T}{\delta}}\|\theta^\star_t - \widehat{\theta}_t\|$. By applying the findings from Theorem 4.1 and the union bound, we show that with probability at least $1 - \delta - \exp(\log T - cN_1) - \exp(d - \frac{c\delta_0^2 N_1 T}{r^2\mu^2\kappa^4}) - 4\exp(\log T + r - cN_1)$, it follows that

$$
\begin{aligned}
|M'_j - M'_{j-1}| &= \sum_{t=1}^{T} x^{\star\top}_{j,t}(\theta^\star_t - \widehat{\theta}_t) + x^\top_{j,t}(\widehat{\theta}_t - \theta^\star_t) \\
&\leqslant 2\sum_{t=1}^{T} \max_{x\in\mathcal{X}} x^\top(\theta^\star_t - \widehat{\theta}_t) \\
&\leqslant 2\sum_{t=1}^{T} \sqrt{2\log\frac{(N-N_1)T}{\delta}}\|\theta^\star_t - \widehat{\theta}_t\| \\
&\leqslant 2\sum_{t=1}^{T} \sqrt{2\log\frac{(N-N_1)T}{\delta}}\left(1.12 + \frac{c}{\kappa^2 r\sqrt{N_1}}\right)\mu\sqrt{\frac{r}{T}}\sigma^\star_{\max}\delta_0 \\
&= 2\sqrt{2}\left(1.12 + \frac{c}{\kappa^2 r\sqrt{N_1}}\right)\mu\sqrt{rT}\sigma^\star_{\max}\delta_0\sqrt{\log\frac{(N-N_1)T}{\delta}}.
\end{aligned}
$$

Utilizing the Azuma-Hoeffding inequality stated in Proposition A.1 and the union bound, we can determine that with probability at least $1 - 2\delta - \exp(\log T - cN_1) - \exp(d - \frac{c\delta_0^2 N_1 T}{r^2\mu^2\kappa^4}) - 4\exp(\log T + r - cN_1)$, the cumulative regret $\mathcal{R}^2_{N,T}$ for the following round is determined as follows:

$$
\begin{aligned}
\mathcal{R}^2_{N,T} &\leqslant \sum_{n=N_1}^{N}\sum_{t=1}^{T} x^{\star\top}_{n,t}(\theta^\star_t - \widehat{\theta}_t) + x^\top_{n,t}(\widehat{\theta}_t - \theta^\star_t) \\
&\leqslant 4\mu\sigma^\star_{\max}\left(1.12 + \frac{c}{\kappa^2 r\sqrt{N_1}}\right)\delta_0\sqrt{r(N-N_1)T\log\frac{1}{\delta}\log\frac{(N-N_1)T}{\delta}}.
\end{aligned}
$$

By combining the bounds for $\mathcal{R}^1_{N,T}$ and $\mathcal{R}^2_{N,T}$ and applying the union bound, we can conclude that with probability at least $1 - 4\delta - \exp(\log T - cN_1) - \exp(d - \frac{c\delta_0^2 N_1 T}{r^2\mu^2\kappa^4}) - 4\exp(\log T + r - cN_1)$,

the cumulative regret $\mathcal{R}_{N,T}$ is given by

$$\mathcal{R}_{N,T} = \mathcal{R}^1_{N,T} + \mathcal{R}^2_{N,T}$$

$$\leqslant 2uT\sqrt{N_1 \log\frac{1}{\delta}\log\frac{N_1 T}{\delta}} + 4\mu\sigma^\star_{\max}\Big(1.12 + \frac{c}{\kappa^2 r\sqrt{N_1}}\Big)\delta_0\sqrt{r(N-N_1)T \log\frac{1}{\delta}\log\frac{(N-N_1)T}{\delta}}$$

$$\leqslant 2uT\sqrt{N \log\frac{1}{\delta}\log\frac{NT}{\delta}} + 4\mu\sigma^\star_{\max}\Big(1.12 + \frac{c}{\kappa^2 r\sqrt{N_1}}\Big)\delta_0\sqrt{rNT \log\frac{1}{\delta}\log\frac{NT}{\delta}}$$

$$= \widetilde{O}\Big(T\sqrt{N} + \delta_0\sigma^\star_{\max}\sqrt{rNT}\Big).$$

Moreover, by combining the cumulative regret bound for $\mathcal{R}^1_{N,T}$, obtained through the $\mu$-column-wise incoherence property of $W^\star$, and $\mathcal{R}^2_{N,T}$, and applying a union bound, we determine that with a probability of at least $1 - 4\delta - \exp(\log T - cN_1) - \exp(d - \frac{c\delta_0^2 N_1 T}{r^2\mu^2\kappa^4}) - 4\exp(\log T + r - cN_1)$, the cumulative regret is bounded by

$$\mathcal{R}N,T \leqslant 2\mu\sigma^\star_{\max}\sqrt{rNT \log\frac{1}{\delta}\log\frac{NT}{\delta}} + 4\mu\sigma^\star_{\max}\Big(1.12 + \frac{c}{\kappa^2 r\sqrt{N_1}}\Big)\delta_0\sqrt{rNT \log\frac{1}{\delta}\log\frac{NT}{\delta}}$$

$$= \Big(2 + 4.48\delta_0 + \frac{4c\delta_0}{\kappa^2 r\sqrt{N_1}}\Big)\mu\sigma^\star_{\max}\sqrt{rNT \log\frac{1}{\delta}\log\frac{NT}{\delta}}$$

$$= \widetilde{O}\Big((1+\delta_0)\sigma^\star_{\max}\sqrt{rNT}\Big)$$

$\square$

# E    Proof of Theorem 5.1

**Definitions:**

For $\lambda > 0$, define the matrices

- $V_{n,T+1} = \lambda I + \widehat{B}^\top \Phi^\top_{n,T+1}\Phi_{n,T+1}\widehat{B}$, and
- $\bar{V}_{n,T+1} = \lambda I + \Phi^\top_{n,T+1}\Phi_{n,T+1}.$

Here $V_{n,T+1} = \widehat{B}^\top \bar{V}_{n,T+1}\widehat{B}.$

**Proof of Theorem 5.1:**

Let $\widehat{w}_{n,T+1}$ be the least squares estimate of $w^\star_{T+1}$ with $\ell^2$ regularization, where the regularization parameter $\lambda > 0$. We have

$$\begin{aligned}
\widehat{w}_{n,T+1} &= V^{-1}_{n,T+1}(\Phi_{n,T+1}\widehat{B})^\top Y_{n,T+1}\\
&= V^{-1}_{n,T+1}(\Phi_{n,T+1}\widehat{B})^\top(\Phi_{n,T+1}B^\star w^\star_{T+1} + H_{n,T+1})\\
&= V^{-1}_{n,T+1}\widehat{B}^\top \Phi^\top_{n,T+1}H_{n,T+1} + V^{-1}_{n,T+1}\widehat{B}^\top \Phi^\top_{n,T+1}\Phi_{n,T+1}B^\star w^\star_{T+1}\\
&= V^{-1}_{n,T+1}\widehat{B}^\top \Phi^\top_{n,T+1}H_{n,T+1} + V^{-1}_{n,T+1}\widehat{B}^\top \Phi^\top_{n,T+1}\Phi_{n,T+1}\widehat{B}\widehat{B}^\top B^\star w^\star_{T+1}\\
&\quad + V^{-1}_{n,T+1}\widehat{B}^\top \Phi^\top_{n,T+1}\Phi_{n,T+1}(I - \widehat{B}\widehat{B}^\top)B^\star w^\star_{T+1}\\
&= V^{-1}_{n,T+1}\widehat{B}^\top \Phi^\top_{n,T+1}H_{n,T+1} + V^{-1}_{n,T+1}(V_{n,T+1} - \lambda I)\widehat{B}^\top B^\star w^\star_{T+1}\\
&\quad + V^{-1}_{n,T+1}\widehat{B}^\top \Phi^\top_{n,T+1}\Phi_{n,T+1}(I - \widehat{B}\widehat{B}^\top)B^\star w^\star_{T+1}\\
&= V^{-1}_{n,T+1}\widehat{B}^\top \Phi^\top_{n,T+1}H_{n,T+1} + \widehat{B}^\top B^\star w^\star_{T+1} - \lambda V^{-1}_{n,T+1}\widehat{B}^\top B^\star w^\star_{T+1}\\
&\quad + V^{-1}_{n,T+1}\widehat{B}^\top \Phi^\top_{n,T+1}\Phi_{n,T+1}(I - \widehat{B}\widehat{B}^\top)B^\star w^\star_{T+1}
\end{aligned}$$

By multiplying $\widehat{B}$ on both sides, we derive

$$
\begin{aligned}
\widehat{B}\widehat{w}_{n,T+1} &= \widehat{B}V_{n,T+1}^{-1}\widehat{B}^\top\Phi_{n,T+1}^\top H_{n,T+1} + \widehat{B}\widehat{B}^\top B^\star w_{T+1}^\star - \lambda\widehat{B}V_{n,T+1}^{-1}\widehat{B}^\top B^\star w_{T+1}^\star \\
&\quad + \widehat{B}V_{n,T+1}^{-1}\widehat{B}^\top\Phi_{n,T+1}^\top\Phi_{n,T+1}(I - \widehat{B}\widehat{B}^\top)B^\star w_{T+1}^\star \\
&= \widehat{B}V_{n,T+1}^{-1}\widehat{B}^\top\Phi_{n,T+1}^\top H_{n,T+1} + B^\star w_{T+1}^\star + (\widehat{B}\widehat{B}^\top - I)B^\star w_{T+1}^\star \\
&\quad - \lambda\widehat{B}V_{n,T+1}^{-1}\widehat{B}^\top B^\star w_{T+1}^\star + \widehat{B}V_{n,T+1}^{-1}\widehat{B}^\top\Phi_{n,T+1}^\top\Phi_{n,T+1}(I - \widehat{B}\widehat{B}^\top)B^\star w_{T+1}^\star
\end{aligned}
$$

Considering any vector $z \in \mathbb{R}^d$, and multiplying both sides, we get

$$
\begin{aligned}
z^\top\widehat{B}&\widehat{w}_{n,T+1} - z^\top B^\star w_{T+1}^\star \\
&= z^\top\widehat{B}V_{n,T+1}^{-1}\widehat{B}^\top\Phi_{n,T+1}^\top H_{n,T+1} - \lambda z^\top\widehat{B}V_{n,T+1}^{-1}\widehat{B}^\top B^\star w_{T+1}^\star + z^\top(\widehat{B}\widehat{B}^\top - I)B^\star w_{T+1}^\star \\
&\quad + z^\top\widehat{B}V_{n,T+1}^{-1}\widehat{B}^\top\Phi_{n,T+1}^\top\Phi_{n,T+1}(I - \widehat{B}\widehat{B}^\top)B^\star w_{T+1}^\star \\
&= \langle(z^\top\widehat{B})^\top, \widehat{B}^\top\Phi_{n,T+1}^\top H_{n,T+1}\rangle_{V_{n,T+1}^{-1}} - \lambda\langle(z^\top\widehat{B})^\top, \widehat{B}^\top B^\star w_{T+1}^\star\rangle_{V_{n,T+1}^{-1}} \\
&\quad + z^\top(\widehat{B}\widehat{B}^\top - I)B^\star w_{T+1}^\star + \langle(z^\top\widehat{B})^\top, \widehat{B}^\top\Phi_{n,T+1}^\top\Phi_{n,T+1}(I - \widehat{B}\widehat{B}^\top)B^\star w_{T+1}^\star\rangle_{V_{n,T+1}^{-1}}
\end{aligned}
$$

Analyzing the absolute value, the upper bound is given as follows:

$$
\begin{aligned}
|z^\top&\widehat{B}\widehat{w}_{n,T+1} - z^\top B^\star w_{T+1}^\star| \\
&\leqslant \|(z^\top\widehat{B})^\top\|_{V_{n,T+1}^{-1}}\|\widehat{B}^\top\Phi_{n,T+1}^\top H_{n,T+1}\|_{V_{n,T+1}^{-1}} + \lambda\|(z^\top\widehat{B})^\top\|_{V_{n,T+1}^{-1}}\|\widehat{B}^\top B^\star w_{T+1}^\star\|_{V_{n,T+1}^{-1}} \\
&\quad + \|(z^\top\widehat{B})^\top\|_{V_{n,T+1}^{-1}}\|\widehat{B}^\top\Phi_{n,T+1}^\top\Phi_{n,T+1}(I - \widehat{B}\widehat{B}^\top)B^\star w_{T+1}^\star\|_{V_{n,T+1}^{-1}} + |z^\top(\widehat{B}\widehat{B}^\top - I)B^\star w_{T+1}^\star| \tag{10}
\end{aligned}
$$

To determine the upper bound for the term $\|\widehat{B}^\top\Phi_{n,T+1}^\top H_{n,T+1}\|_{V_{n,T+1}^{-1}}$, we apply the method from Theorem 1 in [22], which gives us

$$
\|\widehat{B}^\top\Phi_{n,T+1}^\top H_{n,T+1}\|_{V_{n,T+1}^{-1}}^2 \leqslant 2\sigma^2\log\left(\frac{\det(V_{n,T+1})^{\frac{1}{2}}\det(\lambda I)^{-\frac{1}{2}}}{\delta}\right)
$$

with probability at least $1 - \delta$. To determine the upper bound for the term $\|\widehat{B}^\top B^\star w_{T+1}^\star\|_{V_{n,T+1}^{-1}}$, we have

$$
\begin{aligned}
\|\widehat{B}^\top B^\star w_{T+1}^\star\|_{V_{n,T+1}^{-1}}^2 &= w_{T+1}^{\star\top}B^{\star\top}\widehat{B}V_{n,T+1}^{-1}\widehat{B}^\top B^\star w_{T+1}^\star \leqslant \|\widehat{B}^\top B^\star w_{T+1}^\star\|_2^2\|V_{n,T+1}^{-1}\|_2 \\
&= \|\widehat{B}^\top\|_2^2\|B^\star w_{T+1}^\star\|_2^2\|V_{n,T+1}^{-1}\|_2 \leqslant \frac{\|B^\star w_{T+1}^\star\|_2^2}{\lambda_{\min}(V_{n,T+1})} \leqslant \frac{1}{\lambda}S^2
\end{aligned}
$$

To determine the upper bound for the term $\|\widehat{B}^\top\Phi_{n,T+1}^\top\Phi_{n,T+1}(I - \widehat{B}\widehat{B}^\top)B^\star w_{T+1}^\star\|_{V_{n,T+1}^{-1}}$, we have with probability at least $1 - \exp(\log T - cN_1) - \exp(d - \frac{c\delta_0^2 N_1 T}{r^2\mu^2\kappa^4})$,

$$
\begin{aligned}
\|\widehat{B}^\top\Phi_{n,T+1}^\top\Phi_{n,T+1}(I - \widehat{B}\widehat{B}^\top)B^\star w_{T+1}^\star\|_{V_{n,T+1}^{-1}} &\leqslant \|\Phi_{n,T+1}(I - \widehat{B}\widehat{B}^\top)B^\star w_{T+1}^\star\| \\
&\leqslant \|\Phi_{n,T+1}\|\|(I - \widehat{B}\widehat{B}^\top)B^\star\|\|w_{T+1}^\star\| \leqslant \sqrt{n}L\delta_0 S
\end{aligned}
$$

where the first inequality follows from the matrix inequality $\Phi_{n,T+1}\widehat{B}(\lambda I + \widehat{B}^\top\Phi_{n,T+1}^\top\Phi_{n,T+1}\widehat{B})^{-1}\widehat{B}^\top\Phi_{n,T+1}^\top \leqslant I$. The second inequality follows from the Cauchy–Schwarz inequality. The last inequality follows from $\|\Phi_{n,T+1}\| \leqslant \sqrt{n}L$, $\|w_{T+1}^\star\| = \|\theta_{T+1}^\star\| \leqslant S$ and Proposition B.1. Consider $z = \bar{V}_{n,T+1}(\widehat{B}\widehat{w}_{n,T+1} - B^\star w_{T+1}^\star)$. To determine the upper bound for the term

$|z^\top(\widehat{B}\widehat{B}^\top - I)B^\star w^\star_{T+1}|$, we have with probability at least $1 - \exp(\log T - cN_1) - \exp(d - \frac{c\delta_0^2 N_1 T}{r^2\mu^2\kappa^4})$,

$$
\begin{aligned}
|z^\top(\widehat{B}\widehat{B}^\top - I)B^\star w^\star_{T+1}| &= |(\widehat{B}\widehat{w}_{n,T+1} - B^\star w^\star_{T+1})^\top \bar{V}_{n,T+1}(\widehat{B}\widehat{B}^\top - I)B^\star w^\star_{T+1}| \\
&= \langle \widehat{B}\widehat{w}_{n,T+1} - B^\star w^\star_{T+1}, (\widehat{B}\widehat{B}^\top - I)B^\star w^\star_{T+1}\rangle_{\bar{V}_{n,T+1}} \\
&\leqslant \|\widehat{B}\widehat{w}_{n,T+1} - B^\star w^\star_{T+1}\|_{\bar{V}_{n,T+1}} \|(\widehat{B}\widehat{B}^\top - I)B^\star w^\star_{T+1}\|_{\bar{V}_{n,T+1}} \\
&\leqslant \|\widehat{B}\widehat{w}_{n,T+1} - B^\star w^\star_{T+1}\|_{\bar{V}_{n,T+1}}(\sqrt{\lambda}\|(\widehat{B}\widehat{B}^\top - I)B^\star w^\star_{T+1}\| \\
&\quad + \|\Phi_{n,T+1}(\widehat{B}\widehat{B}^\top - I)B^\star w^\star_{T+1}\|) \\
&\leqslant (\sqrt{\lambda} + \sqrt{n}L)\delta_0 S\|\widehat{B}\widehat{w}_{n,T+1} - B^\star w^\star_{T+1}\|_{\bar{V}_{n,T+1}}
\end{aligned}
$$

To determine the upper bound for the term $\|(z^\top\widehat{B})^\top\|^2_{V^{-1}_{n,T+1}}$, we have

$$
\begin{aligned}
\|(z^\top\widehat{B})^\top\|^2_{V^{-1}_{n,T+1}} &= z^\top\widehat{B}V^{-1}_{n,T+1}\widehat{B}^\top z \\
&= (\widehat{B}\widehat{w}_{n,T+1} - B^\star w^\star_{T+1})^\top \bar{V}_{n,T+1}\widehat{B}V^{-1}_{n,T+1}\widehat{B}^\top \bar{V}_{n,T+1}(\widehat{B}\widehat{w}_{n,T+1} - B^\star w^\star_{T+1}) \\
&= (\widehat{B}\widehat{w}_{n,T+1} - B^\star w^\star_{T+1})^\top \bar{V}^{\frac{1}{2}}_{n,T+1}\bar{V}^{\frac{1}{2}}_{n,T+1}\widehat{B}V^{-1}_{n,T+1}\widehat{B}^\top \bar{V}^{\frac{1}{2}}_{n,T+1}\bar{V}^{\frac{1}{2}}_{n,T+1}(\widehat{B}\widehat{w}_{n,T+1} - B^\star w^\star_{T+1}) \\
&\leqslant \|\widehat{B}\widehat{w}_{n,T+1} - B^\star w^\star_{T+1}\|^2_{\bar{V}_{n,T+1}}\|\bar{V}^{\frac{1}{2}}_{n,T+1}\widehat{B}V^{-1}_{n,T+1}\widehat{B}^\top \bar{V}^{\frac{1}{2}}_{n,T+1}\| \\
&\leqslant \|\widehat{B}\widehat{w}_{n,T+1} - B^\star w^\star_{T+1}\|^2_{\bar{V}_{n,T+1}}
\end{aligned}
$$

where the last inequality is derived from $\|\bar{V}^{\frac{1}{2}}_{n,T+1}\widehat{B}V^{-1}_{n,T+1}\widehat{B}^\top \bar{V}^{\frac{1}{2}}_{n,T+1}\| = \|\bar{V}^{\frac{1}{2}}_{n,T+1}\widehat{B}V^{-\frac{1}{2}}_{n,T+1}\|^2 = \lambda_{\max}(V^{-\frac{1}{2}}_{n,T+1}\widehat{B}^\top \bar{V}^{\frac{1}{2}}_{n,T+1}\bar{V}^{\frac{1}{2}}_{n,T+1}\widehat{B}V^{-\frac{1}{2}}_{n,T+1}) = \lambda_{\max}(V^{-\frac{1}{2}}_{n,T+1}V_{n,T+1}V^{-\frac{1}{2}}_{n,T+1}) = 1$. To determine the upper bound for the term $|z^\top\widehat{B}\widehat{w}_{n,T+1} - z^\top B^\star w^\star_{T+1}|$, we have

$$
\begin{aligned}
|z^\top\widehat{B}\widehat{w}_{n,T+1} - z^\top B^\star w^\star_{T+1}| &= |z^\top(\widehat{B}\widehat{w}_{n,T+1} - B^\star w^\star_{T+1})| \\
&= (\widehat{B}\widehat{w}_{n,T+1} - B^\star w^\star_{T+1})^\top \bar{V}_{n,T+1}(\widehat{B}\widehat{w}_{n,T+1} - B^\star w^\star_{T+1}) \\
&= \|\widehat{B}\widehat{w}_{n,T+1} - B^\star w^\star_{T+1}\|^2_{\bar{V}_{n,T+1}}
\end{aligned}
$$

Substituting these in Eq. (10) gives

$$
\begin{aligned}
\|\widehat{B}\widehat{w}_{n,T+1} - B^\star w^\star_{T+1}\|^2_{\bar{V}_{n,T+1}} &\leqslant \|\widehat{B}\widehat{w}_{n,T+1} - B^\star w^\star_{T+1}\|_{\bar{V}_{n,T+1}}\sigma\sqrt{2\log\frac{\det(V_{n,T+1})^{\frac{1}{2}}\det(\lambda I)^{-\frac{1}{2}}}{\delta}} \\
&\quad + \lambda\|\widehat{B}\widehat{w}_{n,T+1} - B^\star w^\star_{T+1}\|_{\bar{V}_{n,T+1}}\frac{1}{\sqrt{\lambda}}S \\
&\quad + \|\widehat{B}\widehat{w}_{n,T+1} - B^\star w^\star_{T+1}\|_{\bar{V}_{n,T+1}}\sqrt{n}L\delta_0 S \\
&\quad + (\sqrt{\lambda} + \sqrt{n}L)\delta_0 S\|\widehat{B}\widehat{w}_{n,T+1} - B^\star w^\star_{T+1}\|_{\bar{V}_{n,T+1}}
\end{aligned}
$$

By rearranging and simplifying the inequality above, we determine that with probability at least $1 - \delta - \exp(\log T - cN_1) - \exp(d - \frac{c\delta_0^2 N_1 T}{r^2\mu^2\kappa^4})$,

$$
\|\widehat{B}\widehat{w}_{n,T+1} - B^\star w^\star_{T+1}\|_{\bar{V}_{n,T+1}} \leqslant \sigma\sqrt{2\log\frac{\det(V_{n,T+1})^{\frac{1}{2}}\det(\lambda I)^{-\frac{1}{2}}}{\delta}} + ((1+\delta_0)\sqrt{\lambda} + 2\sqrt{n}L\delta_0)S.
$$

Furthermore, $\det(V_{n,T+1}) = \det(\lambda I + \widehat{B}^\top \Phi_{n,T+1}^\top \Phi_{n,T+1} \widehat{B}) = \prod_{i=1}^{r}(\lambda + \lambda_i)$, where $\lambda_i$ represent the eigenvalue value of the positive semi-definite matrix $\widehat{B}^\top \Phi_{n,T+1}^\top \Phi_{n,T+1} \widehat{B}$. Given

$$
\begin{aligned}
\lambda_i &\leqslant \mathrm{Tr}(\widehat{B}^\top \Phi_{n,T+1}^\top \Phi_{n,T+1} \widehat{B}) \\
&= \mathrm{Tr}(\sum_{m=1}^{n} \widehat{B}^\top x_{m,T+1} x_{m,T+1}^\top \widehat{B}) \\
&= \sum_{m=1}^{n} \mathrm{Tr}((\widehat{B}^\top x_{m,T+1})(\widehat{B}^\top x_{m,T+1})^\top) \\
&= \sum_{m=1}^{n} (\widehat{B}^\top x_{m,T+1})^\top (\widehat{B}^\top x_{m,T+1}) \\
&= \sum_{m=1}^{n} \|\widehat{B}^\top x_{m,T+1}\|_2^2 \\
&\leqslant \sum_{m=1}^{n} \|x_{m,T+1}\|_2^2 \\
&\leqslant nL^2
\end{aligned}
$$

we conclude that $\det(V_{n,T+1}) \leqslant \prod_{i=1}^{r}(\lambda + nL^2) = (\lambda + nL^2)^r = \lambda^r (1 + \frac{nL^2}{\lambda})^r$. Setting $\delta_0 = \frac{1}{\sqrt{N_2}L}$. Consequently, we show that with probability at least $1 - \delta - \exp(\log T - cN_1) - \exp(d - \frac{c\delta_0^2 N_1 T}{r^2 \mu^2 \kappa^4})$,

$$
\|\widehat{B}\widehat{w}_{n,T+1} - B^\star w_{T+1}^\star\|_{\bar{V}_{n,T+1}} \leqslant \sigma \sqrt{r \log\left(\frac{1 + nL^2/\lambda}{\delta}\right)} + \left(\sqrt{\lambda} + \frac{\sqrt{\lambda}}{\sqrt{N_2}L} + 2\sqrt{\frac{n}{N_2}}\right)S.
$$

Note that we have a probability of $1 - \delta - \exp(\log T - cN_1) - \exp(d - \frac{c\delta_0^2 N_1 T}{r^2 \mu^2 \kappa^4})$. To ensure a probability guarantee of at least $1 - \delta - 2d^{-10}$ for our theorem, it is required to set the bound for $N_1$ and $N_1 T$ such that the exponential terms are less than or equal to $d^{-10}$. We obtain

$$
\begin{aligned}
\log T - cN_1 &\leqslant -10 \log d \Rightarrow N_1 \geqslant C \max(\log d, \log T) \\
d - \frac{c\delta_0^2 N_1 T}{r^2 \mu^2 \kappa^4} &\leqslant -10 \log d \Rightarrow N_1 T \geqslant C\mu^2 \kappa^4 \frac{dr^2}{\delta_0^2}.
\end{aligned}
$$

Consequently, combining these results, we conclude that

$$
\begin{aligned}
N_1 &\geqslant C \max(\log d, \log T) \\
N_1 T &\geqslant C\mu^2 \kappa^4 \frac{dr^2}{\delta_0^2}.
\end{aligned}
$$

By setting $\delta_0 = \frac{1}{\sqrt{N_2}L}$, it is essential to ensure that

$$
\begin{aligned}
N_1 &\geqslant C \max(\log d, \log T) \\
N_1 T &\geqslant C\mu^2 \kappa^4 L^2 dr^2 N_2.
\end{aligned}
$$

Thus, we complete the proof. $\qquad\square$

# F  Proof of Theorem 5.2

In this section we present the proof of Theorem 5.2.

**Proof of Theorem 5.2:**

We start the analysis by bounding the cumulative regret $\mathcal{R}_{N,T+1}$ for the target task $T + 1$. With probability at least $1 - \delta - 2d^{-10}$, we have

$$\mathcal{R}_{N_2,T+1}$$

$$= \sum_{m=1}^{N_2} x_{m,T+1}^{\star\top}\theta_{T+1}^{\star} - x_{m,T+1}^{\top}\theta_{T+1}^{\star}$$

$$\leqslant \sum_{m=1}^{N_2} x_{m,T+1}^{\top}\widetilde{\theta}_{m,T+1} - x_{m,T+1}^{\top}\theta_{T+1}^{\star} \tag{11}$$

$$= \sum_{m=1}^{N_2} x_{m,T+1}^{\top}(\widetilde{\theta}_{m,T+1} - \widehat{\theta}_{m,T+1}) + x_{m,T+1}^{\top}(\widehat{\theta}_{m,T+1} - \theta_{T+1}^{\star})$$

$$\leqslant \sum_{m=1}^{N_2} \left( \|\widetilde{\theta}_{m,T+1} - \widehat{\theta}_{m,T+1}\|_{\bar{V}_{m-1,T+1}} + \|\widehat{\theta}_{m,T+1} - \theta_{T+1}^{\star}\|_{\bar{V}_{m-1,T+1}} \right) \|x_{m,T+1}\|_{\bar{V}_{m-1,T+1}^{-1}}$$

$$\leqslant \sqrt{\sum_{m=1}^{N_2} \left( \|\widetilde{\theta}_{m,T+1} - \widehat{\theta}_{m,T+1}\|_{\bar{V}_{m-1,T+1}} + \|\widehat{\theta}_{m,T+1} - \theta_{T+1}^{\star}\|_{\bar{V}_{m-1,T+1}} \right)^2} \sqrt{\sum_{m=1}^{N_2} \|x_{m,T+1}\|_{\bar{V}_{m-1,T+1}^{-1}}^2} \tag{12}$$

$$\leqslant 2\sqrt{N_2} \left( \sigma\sqrt{r\log\left(\frac{1 + N_2 L^2/\lambda}{\delta}\right)} + (\sqrt{\lambda} + \frac{\sqrt{\lambda}}{\sqrt{N_2}L} + 2)S \right) \sqrt{2\log\left(\frac{\det(\bar{V}_{N_2,T+1})}{\det(\lambda I)}\right)} \tag{13}$$

$$\leqslant 2\sqrt{N_2} \left( \sigma\sqrt{r\log\left(\frac{1 + N_2 L^2/\lambda}{\delta}\right)} + (\sqrt{\lambda} + \frac{\sqrt{\lambda}}{\sqrt{N_2}L} + 2)S \right) \sqrt{2d\log\left(1 + \frac{N_2 L^2}{\lambda}\right)} \tag{14}$$

$$= \widetilde{O}\left(\sqrt{dN_2}\left(\sigma\sqrt{r} + \sqrt{\lambda}S\right)\right),$$

where Eq (11) follows by $x_{m,T+1}^{\star\top}\theta_{T+1}^{\star} \leqslant x_{m,T+1}^{\top}\widetilde{\theta}_{m,T+1}$. Eq (12) follows by Cauchy–Schwarz inequality. Eq (13) follows by Theorem 5.1 and Lemma 11 in [22]. $\qquad\square$

## G   Transferring Feature Representation to New Target Tasks

In this section, we present a direct linear regression algorithm for transfer learning to a new target task using the shared model extracted from the source tasks. Using the estimated linear feature representation $\widehat{B}$ shared across related tasks from the source task, our goal here is to transfer this representation to a new, unseen $(T + 1)^{\text{th}}$ target task and thereby improve learning and sample complexity as compared to the standard approach that learns the task separately without leveraging the knowledge from the source tasks. Our main focus here is to derive a sample complexity bound on the number of target task samples required when using the biased estimate $\widehat{B}$. The pseudocode is presented in Algorithm 3. It uses $\widehat{B}$ as a plug-in surrogate for the unknown $B^{\star}$ and estimates $w_{T+1}^{\star}$. Mathematically, we define our estimator as follows

$$\widehat{w}_{T+1} \in \underset{w\in\mathbb{R}^r}{\arg\min} \sum_{m=1}^{n} \|y_{m,T+1} - x_{m,T+1}^{\top}\widehat{B}w\|^2.$$

In the result below we present the error guarantee for the linear regression estimator and the sample complexity on the number of target samples required.

**Theorem G.1.** *Assume Assumptions 3.1 holds. For new task $T + 1$, with probability at least $1 - \exp(\log T - cN_1) - \exp(d - \frac{c\delta_0^2 N_1 T}{r^2\mu^2\kappa^4}) - 4\exp(\log T + r - cN_2)$, using Algorithm 3, we have*

$$\|\widehat{B}\widehat{w}_{T+1} - B^{\star}w_{T+1}^{\star}\| \leqslant \frac{1}{9}\sigma + 1.12\delta_0\|\theta_{T+1}^{\star}\|$$

---

**Algorithm 3** Transferring Features to New Tasks

---

1: Set number of rounds for target task, $N_2$
2: Perform representation learning using source tasks and estimate $\widehat{B}$ using Algorithm 1 from line 1 to line 10
3: **for** $n \leftarrow 1, \cdots, N_2$ **do**
4:     choose action $x_{n,T+1} = \arg\max_{x \in \mathcal{X}} x^\top \widehat{\theta}_{T+1}$, and obtain $y_{n,T+1}$
5: **end for**
6: Compute $Y_{N_2,T+1} = [y_{1,T+1}, \cdots, y_{N_2,T+1}]^\top$, $\Phi_{N_2,T+1} = [x_{1,T+1}, \cdots, x_{N_2,T+1}]^\top$
7: **Update** $\widehat{w}_{T+1}, \widehat{\theta}_{T+1}$: Set $\widehat{w}_{T+1} \leftarrow (\Phi_{N_2,T+1}\widehat{B})^\dagger Y_{N_2,T+1}$ and set $\widehat{\theta}_{T+1} = \widehat{B}\widehat{w}_{T+1}$

---

*Furthermore, if*

$$N_1 \geqslant C\max(\log d, \log T)$$

$$N_1 T \geqslant C\mu^2\kappa^4 \frac{dr^2}{\delta_0}$$

$$N_2 \geqslant C\max(\log d, \log T, r),$$

*then with probability at least* $1 - 6d^{-10}$,

$$\|\widehat{B}\widehat{w}_{T+1} - B^\star w^\star_{T+1}\| \leqslant \frac{1}{9}\sigma + 1.12\delta_0\|\theta^\star_{T+1}\|$$

*Proof.* Following the same logic as in Lemma C.1, we can demonstrate that with probability at least $1 - 2\exp(\log T + r - cN_2)$, we have

$$\|\widehat{B}(\widehat{B}^\top \Phi_{N_2,T+1}^\top \Phi_{N_2,T+1}\widehat{B})^{-1}\widehat{B}^\top \Phi_{N_2,T+1}^\top H_{N_2,T+1}\| \leqslant \frac{1}{9}\sigma.$$

Following the same logic as in Theorem 4.1, we can show that

$$\widehat{B}\widehat{w}_{T+1} - B^\star w^\star_{T+1} = \widehat{B}(\widehat{B}^\top \Phi_{N_2,T+1}^\top \Phi_{N_2,T+1}\widehat{B})^{-1}\widehat{B}^\top \Phi_{N_2,T+1}^\top H_{N_2,T+1} + (\widehat{B}\widehat{B}^\top - I)B^\star w^\star_{T+1}$$
$$+ \widehat{B}(\widehat{B}^\top \Phi_{N_2,T+1}^\top \Phi_{N_2,T+1}\widehat{B})^{-1}\widehat{B}^\top \Phi_{N_2,T+1}^\top \Phi_{N_2,T+1}(I - \widehat{B}\widehat{B}^\top)B^\star w^\star_{T+1}.$$

Therefore, with probability at least $1 - \exp(\log T - cN_1) - \exp(d - \frac{c\delta_0^2 N_1 T}{r^2\mu^2\kappa^4}) - 4\exp(\log T + r - cN_2)$, we obtain

$$\|\widehat{B}\widehat{w}_{T+1} - B^\star w^\star_{T+1}\|$$
$$\leqslant \|\widehat{B}(\widehat{B}^\top \Phi_{N_2,T+1}^\top \Phi_{N_2,T+1}\widehat{B})^{-1}\widehat{B}^\top \Phi_{N_2,T+1}^\top H_{N_2,T+1}\| + \|(\widehat{B}\widehat{B}^\top - I)B^\star w^\star_{T+1}\|$$
$$+ \|\widehat{B}(\widehat{B}^\top \Phi_{N_2,T+1}^\top \Phi_{N_2,T+1}\widehat{B})^{-1}\widehat{B}^\top \Phi_{N_2,T+1}^\top \Phi_{N_2,T+1}(I - \widehat{B}\widehat{B}^\top)B^\star w^\star_{T+1}\|$$
$$\leqslant \|\widehat{B}(\widehat{B}^\top \Phi_{N_2,T+1}^\top \Phi_{N_2,T+1}\widehat{B})^{-1}\widehat{B}^\top \Phi_{N_2,T+1}^\top H_{N_2,T+1}\|$$
$$+ (1 + 0.12)\|(I - \widehat{B}\widehat{B}^\top)B^\star\|\|w^\star_{T+1}\|$$
$$\leqslant \frac{1}{9}\sigma + 1.12\delta_0\|\theta^\star_{T+1}\|$$

where the second-last inequality is derived from Proposition B.1 in [28]. The last inequality is derived from Proposition B.1. $\qquad\square$

# H Additional Experiments

In this section, we present some additional experiments.

**Comparison with the convex relaxation approach:** We evaluated our proposed algorithm (Algorithm 1) against the convex relaxation method utilizing the same data generation method described in Section 6.1. The convex relaxation approach approximates the non-convex cost function through the application of trace-norm regularization [10]. The key challenge is that the solution to the relaxed problem may not necessarily correspond to a valid solution to the original problem. As shown in

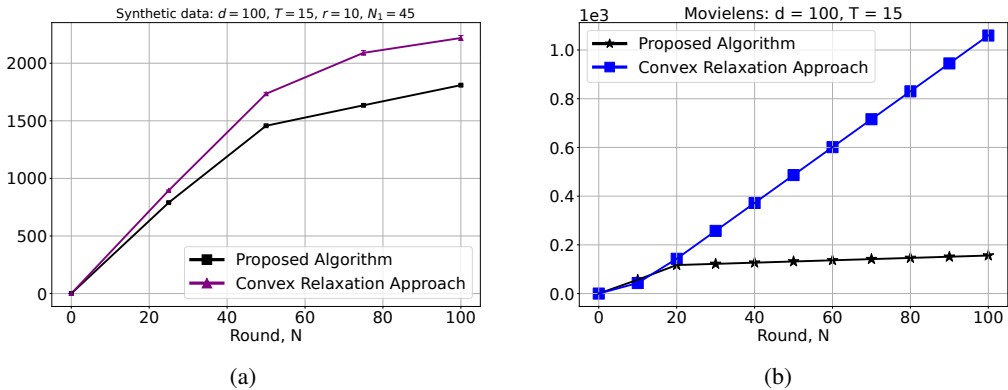

Figure 3: **Plots for synthetic data:** Figure 3a presents the cumulative regret vs. learning round for synthetic data, with parameters set as $d = 100$, $T = 15$, $r = 10$, $N_1 = 45$, $\sigma^2 = 0.01$. **Plots for Movielens data:** Figure 3b presents the cumulative regret vs. learning round for Movielens data, with parameters set as $d = 100$, $T = 15$, $r = 1$, $N_1 = 20$, $\sigma^2 = 0.01$.

Figure 3a, our proposed algorithm outperforms the convex relaxation approach for the synthetic data. Figure 3b indicates that our proposed algorithm shows fast convergence following the exploration phase and consistently outperforms the convex relaxation approach for the Movielens data, which does not converge. A potential reason for this is that the solution to the convex relaxed problem need not necessarily be the solution to the actual non-convex problem. This experiment validates the effectiveness of the proposed approach.

# I  Additional Related Work

We present some additional related work in this section.

**Multi-task reinforcement learning:** Multi-task learning in reinforcement learning (RL) domains is studied in many works, including [36, 39, 55, 41]. [36, 39, 56–58] analyzed the problem from the empirical perspective. From the theoretical perspective, [59] analyzed the sample complexity of multi-task RL in the tabular setting. [55] demonstrated that representation learning has the potential to enhance the rate of the approximate value iteration algorithm. [41] proved that representation learning can reduce the sample complexity of imitation learning. Both works require a probabilistic assumption similar to that in [14] and the statistical rates are of similar forms as those in [14]. Representation learning in multi-task RL has been studied recently in [7, 60–62].

**Low-rank and sparse bandits:** Some previous work studied the impact of low-rank structure in linear bandits [29, 63, 64]. [63] considered a setting where the context vectors consist of two parts, i.e. $\hat{x} = x + \psi$, so that $x$ is from a hidden low-rank subspace and is i.i.d. drawn from an isotropic distribution. The mean reward in [29, 65] is defined as the bilinear multiplication $x^\top \Theta y$, where $x, y$ are the actions chosen and $\Theta$ is the unknown reward matrix with a low-rank structure. The bilinear setting is further generalized by [32].

