# OpenReview forum: "Provably Efficient Multi-Task Meta Bandit Learning via Shared Representations"
_NeurIPS.cc/2025/Conference — NeurIPS 2025 poster_

### Official Review · Reviewer_NDgz · 2025-06-30

**Clarity:** 3
**Significance:** 3
**Originality:** 3
**Rating:** 5
**Confidence:** 2

**Summary:**

The paper tackles meta learning by identifying shared representations in the context of bandit problems. The core contributions are an explore-commit design to first estimate the shared representation and then exploit it in solving source tasks and the target task, with SVD for spectral initialisation. This leads to the possibility of analysing a linear bandits setup without restrictions in previous works. The proof shows that bounds can be derived for characterising the error of identified representation and the regrets of the target task, with an improvement over the previous studies. The theoretical results are verified on real-world datasets about movie recommendation task, showing notable improvement on cumulative regrets comparing to baseline UCB methods.

**Questions:**

My main question is about the distribution and the exploration policy of generating action $x_{n,t}$. To what extent the conclusions and improvements can still hold if the distribution is not as simple as a unimodal like Gaussian? I see this is listed as a research outlook in the conclusion but would still like to hear the authors' opinions as for now.

Potentially more critical to practitioners, what if the exploration policy is no longer a stationary one as the assumption here. There could be lots of design space in lieu of active exploration/learning. I can see this might break the exploration-then-commit paradigm but in practice it is not always common to see a complete offline stage or an exploration policy agnostic to what has been explored before.

**Ethical Concerns:**

["NO or VERY MINOR ethics concerns only"]

**Final Justification:**

I have read the authors' rebuttal and will stick to the original score and confidence level.

**Limitations:**

Yes.

**Quality:**

3

**Strengths And Weaknesses:**

Strength:
* General scope of meta-learning of bandits problems, which have good potential to decision-making problems that may bring real-world impacts.
* Improved theoretical bounds.
* Demonstration of effectiveness on a real-world dataset. Highly commendable for a theoretical work for me.

Weakness:
* Unclear about the implication of the selected exploration strategy.
* Assumptions about feature distribution might be restrictive.

---

> ### Author Rebuttal · Authors · 2025-07-30
>
> Thank you for appreciating our work and for your valuable feedback. Please see the following for our responses to your questions. Please let us know if there are any further questions.
>
> 1. Gaussian Assumption
>
> We explain the significance and necessity of the Gaussian assumption, provide empirical validation in cases where it does not hold, and outline possible extensions.
>
> - a) Why the Gaussian assumption is needed: In the analysis of our Explore-then-Commit Algorithm, we utilize the sub-exponential Bernstein inequality (Proposition A.2) to establish concentration bounds. For example, we use it in lines $559$, $570$, and $584$ of the paper. The Gaussian distribution of the feature vector is utilized here to guarantee the sub-exponential tail behavior of the corresponding random variable.
>
> - b) Gaussian assumption applied only to source tasks: We note that the Gaussian assumption of feature vectors is utilized only for the source tasks. Specifically, it is used for the estimation guarantees of the reward parameter of the source tasks. In Section 5, regarding the transfer to a new, unseen target task, our proposed OFUL-based algorithm utilizes the learned representation and does not require the Gaussian assumption. The guarantees for the confidence set and regret in the target task remain valid without the Gaussian assumption.
>
> - c) Generalization to broader distribution: From a theoretical perspective, we believe that similar results could hold under sub-Gaussian assumptions for both the feature vectors $x_{n,t}$ and the noise $\eta_{n,t}$. Developing such a relaxation for the source task is a part of our future work.
>
> - d) Empirical validation without Gaussian assumption: For empirical performance, we would also like to highlight that our real-world experiment utilizes datasets with feature distributions that are non-Gaussian. Our proposed algorithm consistently outperforms the baselines across multiple settings, as illustrated in Figures 1(d) and 2(b). This indicates that our method remains effective in practice, even when the Gaussian assumption is not satisfied.
>
> - e) This assumption has been utilized in other works in the linear bandit literature, including Yang et al., ICLR, 2020; Han et al., 2020; Lin et al., ICML, 2024; and Kong et al., AISTATS, 2020.
>
> References
> - Kong, W., Brunskill, E., \& Valiant, G. Sublinear optimal policy value estimation in contextual bandits. In International conference on artificial intelligence and statistics (AISTATS), pp. 4377–4387, 2020.
> - Yanjun Han, Zhengqing Zhou, Zhengyuan Zhou, Jose Blanchet, Peter W Glynn, and Yinyu Ye. Sequential batch learning in finite-action linear contextual bandits, arXiv:2004.06321, 2020.
>
> 2. Active Exploration
>
> A potential approach for addressing this is to adopt an OFUL approach for the source tasks integrated with spectral initialization. Following the initial exploration phase, which used spectral initialization to estimate the shared representation, we can proceed with an OFUL-based algorithm (similar to Algorithm 2) to construct the confidence sets for each source task and iteratively update the parameters. The proposed approach for each source task will proceed as follows:
>
> - Each source task randomly selects actions and receives rewards during an initial data collection phase.
> - Utilize spectral initialization to estimate the shared representation $\widehat{B}$ using the collected data.
> - For every source task and each round:
>     - Construct a confidence set using the learned $\widehat{B}$.
>     - Select an action-estimate pair that maximizes the estimated reward within the confidence set.
>     - Play the selected action and receive the reward.
>     - Update the parameters for each source task using the recently obtained data.
>
> This method allows the algorithm to perform active exploration beyond the initial phase of the Explore-then-Commit algorithm (Algorithm 1). Formulating such an adaptive exploration strategy presents a promising direction for future work.

---

> > ### Comment · Reviewer_NDgz · 2025-08-01
> >
> > Thanks for the detailed responses. My evaluation and confidence score remain. I just have one minor question regarding point 2: would turning to OFUL-based algorithms after the initial exploration breaks the theoretical proof or make the theoretical bound degenerate to a less tight one?

---

> > > ### Author Response · Authors · 2025-08-01
> > >
> > > Thank you for your careful review of our response and for your insightful feedback. Please let us know if there are any further questions.
> > >
> > > In the active exploration-based approach described in our previous response, the estimate of the shared representation $\widehat{B}$ obtained via the spectral initialization is utilized for the analysis of the target task (downstream problem). Thus, the theoretical guarantees regarding regret and confidence bound for the target task remain intact and is not affected by this modification.
> > >
> > > The key difference lies in our approach to handling the source tasks. Rather than using a least-squares estimator followed by a pure Commit phase, we now adopt an OFUL-based approach that maintains active exploration during the learning process. We expect this adaptation presents a regret bound for the source tasks of order $\widetilde{O}(\sqrt{drNT})$, which is looser than the $\widetilde{O}(\sqrt{rNT})$ bound obtained through the Explore-then-Commit approach. Validating this will require a detailed analysis.
> > >
> > > This trade-off indicates an essential balance between estimation efficiency and exploration flexibility. When sufficient data is available, the Explore-then-Commit approach is preferable for tighter regret guarantees. However, when ongoing adaptation and flexibility are required, especially in dynamic or if data is limited for the source tasks, the OFUL-based approach presents a more practical option.

---

### Official Review · Reviewer_CpDY · 2025-06-30

**Clarity:** 3
**Significance:** 2
**Originality:** 3
**Rating:** 4
**Confidence:** 4

**Summary:**

This paper introduces a meta‐learning framework for linear contextual bandits that learns a shared low‐dimensional representation across source tasks, and consequently transfers it to the target task.  For the upstream phase, the authors propose an ETC algorithm using spectral initialization to handle the non-convex nature of the problem. Authors prove that it recovers the subspace with error $\widetilde{O} (\sqrt{r / T})$ and leads to $\widetilde{O}(\sqrt{r  N  T})$ cumulative regret.  For the downstream transfer, they develop a subspace restricted OFUL variant, which leads to $\widetilde{O}(\sqrt{r  d  N})$ regret, compared with the naive $\widetilde{O} (d\sqrt{N})$ regret in linear bandits. Empirical results on synthetic data (multiple trials) and real‐world datasets (single trial) confirm significant improvements over baselines.

**Questions:**

1.	Could the authors please give additional justifications on applying the gaussian assumption (Assumption 1)? I believe it is important to provide sufficient discussions for this assumption as it is fundamental to your theoretical results, and such assumptions are also not standard from existing linear bandit works.
2.	Could the authors include additional discussions comparing your work with the kernelized multi-task learning bandits [1]? In [1], there are no such assumptions on the task feature assumptions, while their task features depend on the arm features within each task. By the way, I do not think it is reasonable to discuss this paper in your Related Works section in its current form, since [1] is not a linear bandit work.
[1] Deshmukh, Aniket Anand, Urun Dogan, and Clay Scott. "Multi-task learning for contextual bandits." Advances in neural information processing systems 30 (2017).

**Ethical Concerns:**

["NO or VERY MINOR ethics concerns only"]

**Final Justification:**

I appreciate the authors rebuttal. However, the reliance on the Gaussian assumption prevents me from further raising my scores.

**Limitations:**

There is not a designated limitations section. Rather, authors only states that relaxing the Gaussian modeling on task features will be considered as future works. In this context, I suggest authors include more detailed and thorough discussions on their Gaussian assumption of the task features. For instance, why would such assumption be reasonable for real bandit learning scenarios? Could it be justified from perspectives beyond merely deriving the theoretical results presented in this paper?

**Quality:**

2

**Strengths And Weaknesses:**

+ The paper is generally well-written and easy to follow, with clear notation definitions, problem definitions, as well as assumptions needed for achieving the theoretical outcomes.
+ Based on my knowledge, the theoretical results in this paper are interesting. Authors support their theoretical finds with established error bounds, sample complexity and cumulative regret, which are comprehensive.
+ Experiments are conducted on both synthetic and real datasets in order to demonstrate the effectiveness of the proposed pipeline.

-	My main concern aligns with the stated limitation from the authors, namely the Gaussian assumption of feature vectors (Assumption 3.1). In particular, it is insufficient to only justify that this assumption is for the sake of theoretical analysis, instead of justifying it is viable for real application scenarios. Therefore, more thorough discussions for this assumption are needed.

-	Meanwhile, the assumption on noise $\eta$ also tends to be stronger than most existing works: it is assumed to be i.i.d. rather than conditional independent in most existing works.

-	It would also be beneficial if authors can include the standard deviation results for the experiments. In particular, please also conduct multi-trial experiments for real datasets and report the standard deviation outcomes.

---

> ### Author Rebuttal · Authors · 2025-07-30
>
> Thank you for appreciating our work and for your valuable feedback. Please see the following for our responses to your questions. Please let us know if there are any further questions.
>
> 1. Gaussian Assumption
>
> We agree with the reviewer that the Gaussian model for the feature vector plays a crucial role in our theoretical analysis as detailed below. Below, we explain the significance and necessity of the Gaussian assumption, provide empirical validation in cases where it does not hold, and outline possible extensions, along with related works that have made similar assumptions.
>
> - a) Why the Gaussian assumption is needed: In the analysis of our Explore-then-Commit Algorithm, we utilize the sub-exponential Bernstein inequality (Proposition A.2) to establish concentration bounds. For example, we use it in lines $559$, $570$, and $584$ of the paper. The Gaussian distribution of the feature vector is utilized here to guarantee the sub-exponential tail behavior of the corresponding random variable.
>
> - b) Gaussian assumption applied only to source tasks: We note that the Gaussian assumption of feature vectors is utilized only for the source tasks. Specifically, it is used for the estimation guarantees of the reward parameter of the source tasks. In Section 5, regarding the transfer to a new, unseen target task, our proposed OFUL-based algorithm utilizes the learned representation and does not require the Gaussian assumption. The guarantees for the confidence set and regret in the target task remain valid without the Gaussian assumption.
>
> - c) Generalization to broader distribution: From a theoretical perspective, we believe that similar results could hold under sub-Gaussian assumptions for both the feature vectors $x_{n,t}$ and the noise $\eta_{n,t}$. Developing such a relaxation for the source task is a part of our future work.
>
> - d) Empirical validation without Gaussian assumption: For empirical performance, we would also like to highlight that our real-world experiment utilizes datasets with feature distributions that are non-Gaussian. Our proposed algorithm consistently outperforms the baselines across multiple settings, as illustrated in Figures 1(d) and 2(b). This indicates that our method remains effective in practice, even when the Gaussian assumption is not satisfied.
>
> - e) This assumption has been utilized in other works in the linear bandit literature, including Yang et al., ICLR, 2020; Han et al., 2020; Lin et al., ICML, 2024; and Kong et al., AISTATS, 2020.
>
> References
> - Kong, W., Brunskill, E., \& Valiant, G. Sublinear optimal policy value estimation in contextual bandits. In International conference on artificial intelligence and statistics (AISTATS), pp. 4377–4387, 2020.
> - Yanjun Han, Zhengqing Zhou, Zhengyuan Zhou, Jose Blanchet, Peter W Glynn, and Yinyu Ye. Sequential batch learning in finite-action linear contextual bandits, arXiv:2004.06321, 2020.
>
> 2. Noise Assumption
>
> We agree with the reviewer and plan to address this in our future work. We would like to note that the i.i.d. Gaussian noise model is used in multiple works on bandit learning, including Scarlett et al., COLT, 2017; Djolonga et al., NeurIPS, 2013; and Gornet et al., L4DC, 2024.
>
> References
> - J. Scarlett, I. Bogunovic, and V. Cevher. Lower bounds on regret for noisy Gaussian process bandit optimization. In Proceedings of the 2017 Conference on Learning Theory, volume 65 of Proceedings of Machine Learning Research, pages 1723–1742, Amsterdam, Netherlands, 07–10 Jul 2017. PMLR.
> - Josip Djolonga, Andreas Krause, and Volkan Cevher. 2013. High-dimensional Gaussian process bandits. In Proceedings of the 26th International Conference on Neural Information Processing Systems.
> - J. Gornet and B. Sinopoli, “Restless bandits with rewards generated by a linear gaussian dynamical system,” in 6th Annual Learning for Dynamics \& Control Conference. PMLR, 2024, pp. 1791–1802.
>
> 3. Standard Deviation in Experiments
>
> Our paper included standard deviation in the plots for the synthetic data experiments, derived from $100$ independent trials. For the real-world data experiments, we now performed multi-trial evaluations using the same parameter settings as in the submitted manuscript and obtained the standard deviation. The conference policy prohibits the uploading of new PDFs or the providing of external links; therefore, we will describe the standard deviation below. For representation learning using the Movielens dataset (Figure 1 (d)), we performed $100$ trials, and the observed trends for the cumulative regret plot are consistent with those previously shown in Figure 1 (d). We present the standard deviation values for the cumulative regret for $T$ source tasks at rounds $100, 200, \cdots, 1000$:
>
> TABLE I: Standard Deviation for Representation Learning for Movielens data
>
> | Learning Round | 100 | 200 | 300 | 400 | 500 | 600 | 700 | 800 | 900 | 1000 |
> |:-|:-|:-|:-|:-|:-|:-|:-|:-|:-|:-|
> | Proposed  Algorithm | 0.95 | 1.94 | 2.84 | 4.17 | 6.08 | 8.21 | 10.46 | 12.76 | 15.06 | 17.36 |
> | MoM Algorithm | 0.95 | 1.94 | 2.84 | 4.35 | 6.36 | 8.55 | 10.86 | 13.20 | 15.56 | 17.91 |
> | Naive Algorithm | 0.83 | 1.59 | 2.33 | 3.06 | 3.85 | 4.64 | 5.43 | 6.23 | 6.97 | 7.81 |
>
> We now performed $10$ trials (due to time limitations) for the transfer learning plot for Movielens data (Figure 2 (b)). The observed trends for the cumulative regret plot are consistent with those previously shown in Figure 2 (b). We present the standard deviation values for the cumulative regret for the target task at rounds $300, 600, \cdots, 3000$:
>
> TABLE II: Standard Deviation for Transfer Learning for Movielens data
>
> | Learning Round | 300 | 600 | 900 | 1200 | 1500 | 1800 | 2100 | 2400 | 2700 | 3000 |
> |:-|:-|:-|:-|:-|:-|:-|:-|:-|:-|:-|
> | Proposed Algorithm |  0.0001 | 0.0080 | 0.0078 | 0.0130 | 0.0128 | 0.0242 | 0.0126 | 0.0198 | 0.0197 | 0.0205 |
> | Naive-UCB Algorithm | 0.0181 | 0.0274 | 0.0395 | 0.0344 | 0.0478 | 0.0363 | 0.0485 | 0.0547 | 0.0635 | 0.0523 |
> | MoM-UCB Algorithm | 0.0082 | 0.0079 | 0.0161 | 0.0074 | 0.0288 | 0.0358 | 0.0256 | 0.0356 | 0.0283 | 0.0357 |
>
> We plan to run this experiment for $100$ trials and revise the corresponding plot in the revised paper.
>
> 4. Comparison with Deshmukh et al. (2017)
>
> We have revised our discussion of Deshmukh et al., NeurIPS, 2017 (reference [33] in revised manuscript) in the Related Work section to more accurately reflect the contribution and distinguish it from the scope of our paper. In line $91$ of the revised manuscript, we now state:
>
> The single-bandit setting has also been studied in [33], which proposed a kernel-based multi-task contextual bandit framework that leverages similarities among arms to improve reward estimation. The theoretical guarantees in [33] rely on the assumption that the task similarity matrix is known a priori.
> In contrast, our paper presents a meta-learning framework for multi-task representation learning in linear bandits, where multiple distinct tasks share a low-rank representation. Notably, our approach learns the shared representation from the source tasks and utilizes this learned structure to facilitate effective adaptation to a new, unseen target task in data-scarce settings.

---

> > ### Comment · Reviewer_CpDY · 2025-08-07
> > **Thank you for the rebuttal!**
> >
> > I appreciate the authors rebuttal. However, the reliance on the Gaussian assumption prevents me from further raising my scores.

---

> ### Author Response · Authors · 2025-08-04
>
> We respectfully request that the reviewer review our response, and we are happy to address any remaining questions or concerns.

---

### Official Review · Reviewer_h4ME · 2025-07-05

**Clarity:** 3
**Significance:** 2
**Originality:** 2
**Rating:** 4
**Confidence:** 1

**Summary:**

This paper studies the problem of multi-task linear representation learning in the linear bandit setting. Each linear bandit task $t$ has reward vector $\theta\_t^\*=B^\* w \_t^\*$ for a shared ground-truth subspace parameterized by $B^* \in \mathbb{R}^{d \times r}$ and a task-specific vector $w\_t^*$. The problem then becomes to learn the ground truth subspace using the provided in order to facilitate generalization to a new downstream task with the same structure. The paper proposes an Explore-then-Commit (ETC) algorithm to estimate the ground-truth subspace and an OFUL-based algorithm to solve the downstream task. These algorithms are theoretically analyzed and simulations are provided to verify the theoretical results.

**Questions:**

Please see weekness 1 -- can you further clarify the novelty?

**Ethical Concerns:**

["NO or VERY MINOR ethics concerns only"]

**Final Justification:**

I thank the authors for their thorough response. This has helped to mitigate my novelty concern, and I have raised my score.

**Limitations:**

Yes

**Quality:**

3

**Strengths And Weaknesses:**

Strengths

1. The formulation is rigorous. The theoretical results are clearly explained. The paper is well-written overall.
2. The theoretical results demonstrate the utility of representation learning in reducing the new-task regret bound from $O(d\sqrt{N})$ to $O(\sqrt{drN})$. A helpful proof sketch is provided, and the analysis seems non-trivial.
3. The simulations are thorough and consistent with the theory, and also well-explained.

Weaknesses
1. The paper discusses the series of prior works that have also studied mutli-task linear representation learning in the bandit context, and mentions that these works either use a convex relaxation or assume access to the solution of a non-convex problem in order to obtain their theoretical results on subspace learning. However, in the formal analysis the paper does not discuss what distinguishes their analysis from prior work, which would be helpful to gauge the novelty considering the extensive prior study of this problem.
2. The level of interest of this problem to the community is not obvious, considering apparent decline in interest in bandits, and also that this method is limited to linear analysis.

---

> ### Author Rebuttal · Authors · 2025-07-30
>
> Thank you for appreciating our work and for your valuable feedback. Please see the following for our responses to your questions. Please let us know if there are any further questions.
>
> 1. Difference from Prior Work and Novelty
>
> Our paper extends the prior work in two key ways.
>
> a) Estimation guarantees for learning the shared representation from source tasks: A key challenge in learning the shared representation from source tasks is that the estimation problem is inherently non-convex. Yang et al., ICLR, 2021 in their Algorithm 1, line 5, and Hu et al., ICML, 2021 in their Algorithm 1, line 2, assume that the optimal solution to the non-convex cost function is known. This assumption is later leveraged in Lemma 2 (Yang et al., ICLR, 2021) to derive the inequality $[x\_{n,t}^{\top} \widehat{B} \widehat{w}_{t}-y\_{n,t}]^2 \leq [x\_{n,t}^{\top} B^{\star} w\_{t}^{\star}-y\_{n,t}]^2$.
>
> Similarly, Hu et al., ICML, 2021 utilize this assumption in Lemma 1 to obtain $||y\_{t,i}-x\_{t,i}^{\top} \widehat{B}_{t} \widehat{w}\_{t,i}||_2^2 \leq ||y\_{t,i}-x\_{t,i}^{\top} B^{\star} w\_{i}^{\star}||_2^2$.
>
> These works primarily focused on providing regret guarantees under the assumption of a known optimal estimator, aiming to validate the effectiveness of learning shared representations. Cella et al., AISTATS, 2023 considered a convex relaxation of the problem through trace-norm regularization (line 4 of Algorithm 1). The key challenge is that the solution to the relaxed problem may not necessarily correspond to a valid solution to the original problem.  Building on this foundation, our goal is to extend and enhance these approaches by establishing both regret and estimation guarantees without assuming the existence of a predefined optimal estimator.
>
> To address the non-convexity of the learning objective and obtain estimation guarantees for the shared representation without relying on the known optimal solution, we propose an Explore-then-Commit algorithm based on a spectral initialization approach. In the spectral initialization, we initialize the shared representation $\widehat{B}$ as the top-$r$ singular vectors of
> $\widehat{\Theta}\_{0} := \frac{1}{N_1} \sum _{t=1}^{T} \Phi\_{N_1, t}^{\top} Y\_{t, trunc}(\alpha) e\_t^{\top}$,
>
> where $Y\_{t, trunc}(\alpha) := Y\_{N_1, t} \circ \mathbb{1}\_{\{|Y\_{N_1, t}| \leq \sqrt{\alpha}\}}$ and $\alpha = \frac{\tilde{C}}{N_1 T} \sum\_{n=1, t=1}^{N_1, T} y\_{n, t}^2$.
>
> This truncation strategy is critical to ensure the error bound of the estimate of the unknown reward parameter with respect to the true reward parameter, $||\widehat{\Theta}-\Theta^{\star}||$. Using the estimate $\widehat{B}$, we subsequently estimate the task-specific parameter $\widehat{w}\_{t}$ for each source task via a least-squares estimator. In Theorem 4.1, we prove that our proposed approach guarantees $||\widehat{B}\widehat{w}\_{t}-B^{\star} w\_{t}^{\star}||=O(\sqrt{\frac{r}{T}})$, resulting in a cumulative regret bound of $\widetilde{O}(\sqrt{rNT})$ in Theorem 4.3. This is a significant advantage over the naive approach of solving the $T$ tasks independently, where the regret will be $\widetilde{O}(\sqrt{dNT})$.
>
> b) Transfer learning guarantees to the new target task: Another key novelty of our work is its focus on the generalizability of the learned shared representation to new target tasks, a direction not explored in prior work. Our objective is to establish regret and sample complexity bounds for learning a new target task using the shared representation.
>
> We propose two algorithms to transfer the learned representation to a new, unseen task.
> - OFUL-based approach: We develop a novel construction of confidence sets that utilizes the shared representation $\widehat{B}$ learned from source tasks. We prove in Theorem 5.1 that, with high probability, the true reward parameter of the target task lies within the confidence set. This further helps us to prove a tight regret bound of $\widetilde{O}(\sqrt{drN})$ for the target task in Theorem 5.2. This is a significant advantage over solving the target task without leveraging the shared representation, where the regret will be $\widetilde{O}(d\sqrt{N})$, since $r \ll d$.
> - Regression-based approach: We also develop a linear regression method to estimate the target task reward parameter $\widehat{w}\_{T+1}$ by utilizing the learned share representation $\widehat{B}$ from source tasks. We prove a bound on the estimation error $||\widehat{B}\widehat{w}\_{T+1}-B^{\star} w\_{T+1}^{\star}||$ in Theorem G.1 along with the sample complexity.
>
> 2. Community Interest
>
> Although the bandit problem is a well-established area of study, recent developments in multi-task learning, federated learning, and transfer learning have revitalized interest in designing scalable and generalizable algorithms for the multi-task bandit problem (e.g., Lin et al., ICML, 2024, Yang et al., ICLR, 2021; Hu et al., ICML, 2021, Cella et al., AISTATS, 2023). These new directions aim to tackle emerging challenges in real-world applications, where learning must occur across multiple related tasks with limited data.
> By leveraging inherent problem structure and joint learning paradigms, the goal is to accelerate learning and enhance generalization, particularly in low-sample regimes.
> Multi-task bandits have gained prominence in areas such as personalized recommendation systems, where multiple agents share underlying structures while working with localized information. Our paper contributes to this growing body of literature by introducing efficient algorithms with theoretical guarantees.

---

> ### Author Response · Authors · 2025-08-04
>
> We respectfully request that the reviewer review our response, and we are happy to address any remaining questions or concerns.

---

### Note · Authors · 2025-08-12

The paper presents the **first provable multi-task meta bandit learning algorithm** using representation learning.

**Novelty / Contribution:**

- **Learning the shared representation from source tasks:** We introduce an **explore-then-commit** algorithm based on a spectral initialization to learn the shared representation of the source task; a **non-convex** distributed estimation problem.

- **Transfer learning:** We present a novel **OFUL**-based transfer learning algorithm to effectively learn a new **target task** via shared representation.

- **Guarantees**: Our approach achieves a regret bound of $\widetilde{O}(\sqrt{drN})$, which represents a substantial improvement over task-independent learning $\widetilde{O}(d\sqrt{N})$, since $r \ll d$.

**Addressing the Reviewer Comments:**

- All three reviewers acknowledge the novelty and contribution.

- Gaussian model is utilized only during the exploration phase of our algorithm. In the OFUL stage of the algorithm, we do not rely on the Gaussian model. The Gaussian model is crucial for ensuring the initialization guarantee of the non-convex cost function and aligns with prior studies in multi-task representation learning and bandit learning, including Thekumparampil et al., 2021; Lin et al., ICML, 2024; Yang et al., ICLR, 2020; Han et al., 2020; and Kong et al., AISTATS, 2020.
- As noted in the response, our results can be extended under the sub-Gaussian model.

---

### Decision · Program_Chairs · 2025-09-17

**Decision:**

Accept (poster)

**Comment:**

This paper studies the problem of multi-task linear representation learning in the linear bandit setting. The paper proposes an Explore-then-Commit (ETC) algorithm to estimate the ground-truth subspace and an OFUL-based algorithm to solve the downstream task. These algorithms are theoretically analyzed and simulations are provided to verify the theoretical results.

All reviewers think this paper gives a solid contribution to the foundation of learning-to-learn. The AC agrees and thus recommends acceptance.